EMBO
reports

# Novel role of bone morphogenetic protein 9 in innate host responses to HCMV infection

Markus Stempel [iD] [1,2], Oliver Maier [iD] [1], Baxolele Mhlekude [iD] [2], Hal Drakesmith [iD] [3] & Melanie M Brinkmann [iD] [1,2 ✉]

## Abstract

Herpesviruses modulate immune control to secure lifelong infection. The mechanisms Human Cytomegalovirus (HCMV) employs in this regard can reveal unanticipated aspects of cellular signaling involved in antiviral immunity. Here, we describe a novel relationship between the TGF-β family cytokine BMP9 and HCMV infection. We identify a cross-talk between BMP9-induced and IFN receptor-mediated signaling, showing that BMP9 boosts the transcriptional response to and antiviral activity of IFNβ, thereby enhancing viral restriction. We also show that BMP9 is secreted by human fibroblasts upon HCMV infection. However, HCMV infection impairs BMP9-induced enhancement of the IFNβ response, indicating that this signaling role of BMP9 is actively targeted by HCMV. Indeed, transmembrane proteins US18 and US20, which downregulate type I BMP receptors, are necessary and sufficient to cause inhibition of BMP9-mediated boosting of the antiviral response to IFNβ. HCMV lacking US18 and US20 is more sensitive to IFNβ. Thus, HCMV has a mutually antagonistic relationship with BMP9, which extends the growing body of evidence that BMP signaling is an underappreciated modulator of innate immunity in response to viral infection.

Keywords Bone Morphogenetic Protein (BMP); BMP9; Human Cytomegalovirus (HCMV); US18; US20
Subject Categories Immunology; Microbiology, Virology & Host Pathogen Interaction; Signal Transduction

## Introduction

The opportunistic herpesvirus Human cytomegalovirus (HCMV) can cause severe clinical complications in immunosuppressed patients, and is the leading viral cause of birth defects (Cohen and Corey, 1985; Meyers et al, 1986; Ramsay et al, 1991). Treatment options are limited, and a licensed HCMV vaccine does not exist. To establish a chronic, lifelong infection, herpesviruses need to modulate the immune response of their host. Studying viral immune evasion has revealed important insights into antiviral cellular restriction mechanisms, improving our understanding of the complex relationship between host and pathogen (Bowie and Unterholzner, 2008; Eaglesham and Kranzusch, 2020; Fabits et al, 2020; Stempel et al, 2019; Zhang et al, 2022).

The bone morphogenetic protein (BMP) and Activin signaling pathways have recently been implicated as new regulators of antiviral innate immune responses (Eddowes et al, 2019; Jiyarom et al, 2022; Zhong et al, 2021). BMPs and Activins belong to the transforming growth factor β (TGF-β) family of cytokines, and are multi-functional growth factors (Ganjoo et al, 2022; Nickel and Mueller, 2019). BMP ligands bind specific receptor complexes, consisting of one type I and one type II receptor, subsequently inducing the formation of an intracellular trimeric transcription factor complex, consisting of Mothers against decapentaplegic homolog (SMAD) proteins, resulting in the transcriptional activation of SMAD-responsive genes (Fig. 1). Activation of both type I interferon (IFN) receptor (IFNAR) signaling, leading to the expression of interferon-stimulated genes (ISGs) and an antiviral cellular state, and BMP signaling is controlled by negative regulators to either prevent overactivation, limit inappropriate activation, or terminate signaling (Lemmon et al, 2016). A well-studied negative regulator of IFNAR signaling is the ubiquitin-specific protease 18 (USP18), which is induced by IFN signaling and is recruited to the IFNAR2 receptor subunit to terminate signaling (Arimoto et al, 2017; Malakhova et al, 2006); interestingly, USP18 expression is decreased by BMP6 (Eddowes et al, 2019). The SMAD-specific E3 ubiquitin protein ligase 1 (Smurf1) negatively regulates BMP-mediated signaling by ubiquitinating SMAD proteins and BMP receptors, subsequently preventing further signaling activation (Murakami and Etlinger, 2019; Zhu et al, 1999). Notably, Smurf1 was also found to inhibit IFN-mediated signaling by initiating STAT1 ubiquitination and degradation (Yuan et al, 2012). These links of USP18 and Smurf1 to both, IFNAR- and BMP-mediated signaling, indicate that these two signaling pathways may be tightly interconnected during the antiviral immune response (Fig. 1).

So far, only very few studies investigated the role of BMPs during viral infection. Aside from the antiviral role of BMP6 during HCV infection, and BMP6 and Activin A during HBV and Zika

[1]Institute of Genetics, Technische Universität Braunschweig, Braunschweig, Germany. [2]Virology and Innate Immunity Research Group, Helmholtz Centre for Infection Research, Braunschweig, Germany. [3]MRC Translational Immune Discovery Unit, MRC Weatherall Institute of Molecular Medicine, University of Oxford, John Radcliffe Hospital, Oxford, UK. ✉E-mail: m.brinkmann@tu-braunschweig.de

   

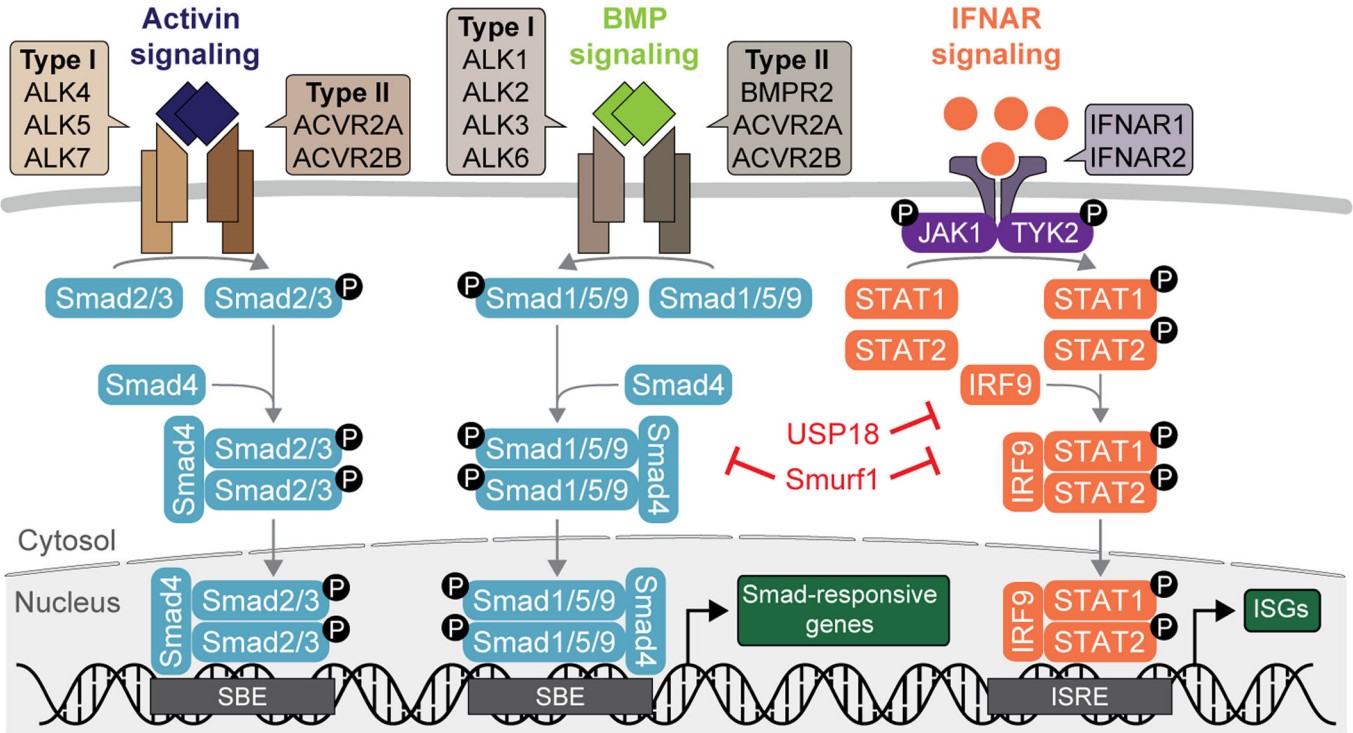

**Figure 1. Canonical Activin-, BMP-, and IFN-mediated signaling pathways.**

Activins and bone morphogenetic proteins (BMPs) bind to ligand-specific type I and type II receptors on the cell surface, upon which an intracellular signaling cascade via Suppressor of Mothers Against Decapentaplegic homolog (SMAD) 2/3 (Activin signaling) or SMAD1/5/9 (BMP signaling) proteins is induced. Dimerized SMAD2/3 or SMAD1/5/9 proteins, respectively, form a trimeric complex with SMAD4, translocate to the nucleus and bind to SMAD-binding elements (SBE) in the genome to activate transcription of Smad-responsive genes. Type I interferons (IFNs) bind to the type I IFN receptor (IFNAR1/2), inducing signaling via the signal transducer and activator of transcription (STAT) 1/2 proteins, in concert with interferon-regulatory factor 9 (IRF9), leading to the transcription of interferon-stimulated gene products (ISGs). The ubiquitin-specific peptidase 18 (USP18) is an IFNAR signaling repressor, while the SMAD-specific E3 ubiquitin protein ligase 1 (Smurf1) negatively regulates SMAD and IFNAR signaling.

virus infection (Eddowes et al, 2019; Jiyarom et al, 2022), BMP8a was shown to be a positive regulator of antiviral immunity upon RNA virus infection in zebrafish by modulating pattern recognition receptor (PRR) signaling leading to type I interferon (IFN) production (Zhong et al, 2021). Recently, the BMP receptor BMPR2 was reported to be crucial for the establishment of HCMV latency (Poole et al, 2021), highlighting the importance of BMP-mediated signaling for infection outcome. Aside from BMPs, TGFβ was recently shown to be induced upon lytic infection and limit induction of type I IFN (Pham et al, 2021). However, the direct impact of BMP-mediated signaling on herpesviral replication during the acute phase of infection has not been characterized yet. Further evidence pinpointing to a role of BMP signaling for HCMV originates from a proteomics screen, where two HCMV proteins, US18 and US20, were identified to downregulate BMP receptors from the cell surface during HCMV infection (Fielding et al, 2017), although the consequences of this effect, or BMP-HCMV interactions more generally, were not explored.

Here, we show that stimulation of human fibroblasts (HFF-1) with BMP9 enhances the antiviral activity of type I IFN on HCMV infection, restraining viral replication. Furthermore, we show that BMP9 is secreted by HFF-1 upon HCMV infection, but that HCMV US18 and US20 specifically inhibit BMP9-mediated signaling during HCMV infection, thus preventing the enhanced BMP9-

induced cellular host response. An HCMV mutant lacking US18 and US20 expression is more sensitive to treatment with IFNβ at early stages of infection, highlighting an involvement of BMP9 in a strong antiviral immune response to HCMV infection.

# Results

## BMPs induce transcription of interferon-stimulated genes in human fibroblasts

To test if BMP signaling has antiviral activity during acute HCMV infection, we first verified the expression of BMP receptors and their responsiveness to BMP stimulation in human foreskin fibroblasts (HFF-1), which are permissive for HCMV. HFF-1 express transcripts of the BMP type I (ALK1, ALK2, ALK3, ALK6) and type II (BMPR2, ACVR2A, ACVR2B) receptors (Fig. 2A), and endogenous ALK1, ALK2, ALK3, and BMPR2 protein expression was also verified by immunoblotting (Fig. EV1A). For further assays, since BMPs or BMP-like factors are present in fetal bovine serum (FBS) (Kodaira et al, 2006), we serum-starved HFF-1 for 2 h prior to treatment to minimize background levels of stimulation (Fig. 2B). Next, we stimulated HFF-1 with the recombinant BMPs BMP4, BMP6, BMP9, BMP15, and Activin B as representative

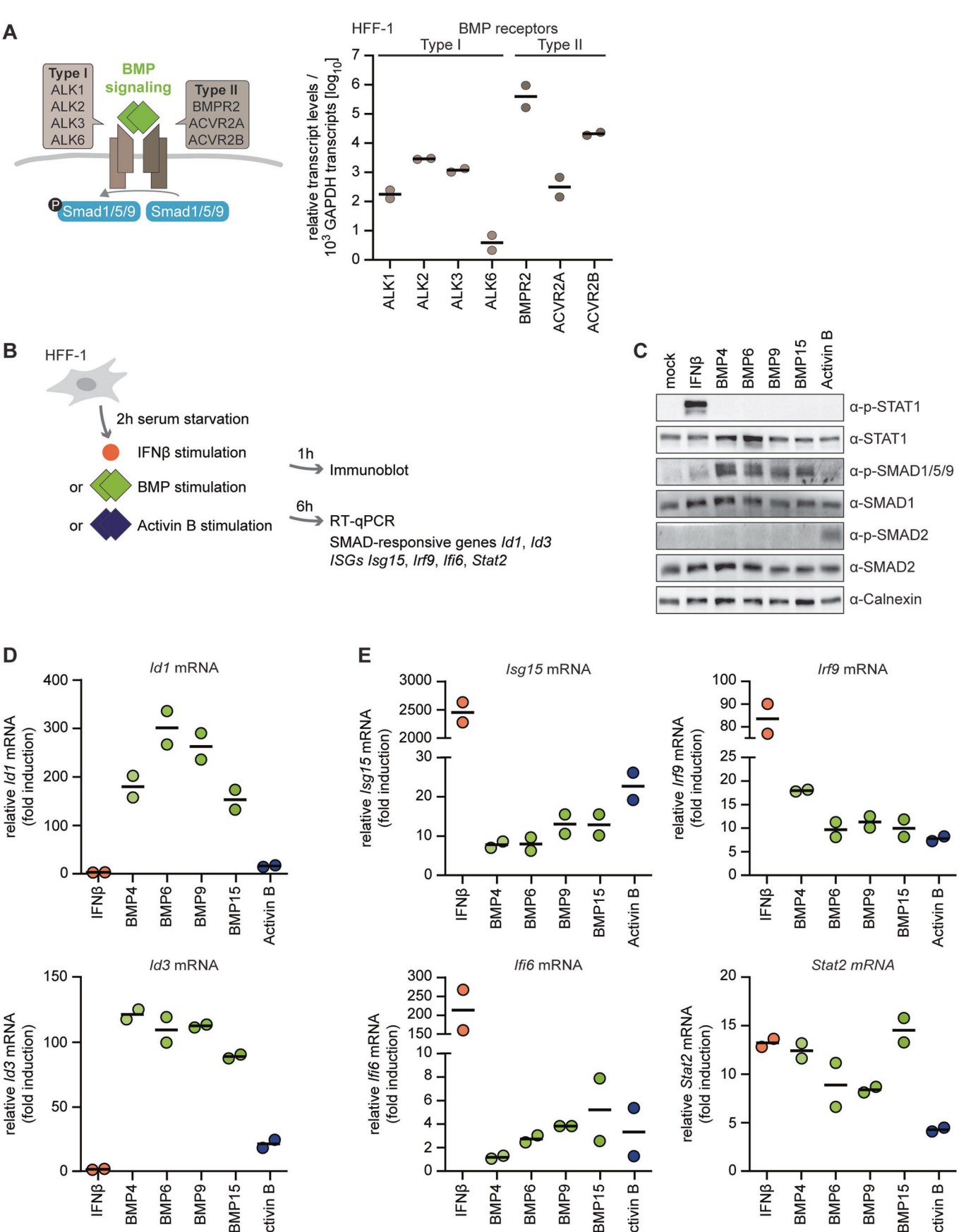

**Figure 2. Stimulation of HFF-1 with BMPs leads to transcriptional activation of ISGs.**

(A) Relative transcript levels of type I (ALK1, ALK2, ALK3, ALK6) and type II (BMPR2, ACVR2A, ACVR2B) BMP receptors in primary human foreskin fibroblasts (HFF-1) were determined by RT-qPCR. (B) Schematic representation of the workflow. After 2 h of serum starvation, HFF-1 were stimulated with IFNβ (5 ng/ml), BMP4 (18 nM), BMP6 (18 nM), BMP9 (3 nM), BMP15 (18 nM), or Activin B (4 nM), followed by either immunoblot analysis 1 h post stimulation or RNA extraction from cell lysates and RT-qPCR 6 h post stimulation. GAPDH transcript levels were used for normalization. (C) Immunoblot analysis of HFF-1 upon stimulation to determine phosphorylation levels of respective signaling components. (D) Transcript levels of the BMP-responsive genes *Id1* and *Id3* upon stimulation with the indicated ligands. (E) Transcript levels of the ISGs *Isg15*, *Irf9*, *Ifi6*, and *Stat2* upon stimulation with the indicated ligands. Data information: (A, B) The Experiment was performed two independent times, one representative is shown. (D, E) The experiment was performed three independent times, one representative is shown. Data are shown as mean ± SD. Source data are available online for this figure.

members of the BMP/Activin family based on their divergent phylogeny, and show that all tested BMPs induced phosphorylation of SMAD1/5/9, and Activin B induced phosphorylation of SMAD2, whereas BMPs and Activin B did not cause phosphorylation of STAT1, which was induced by IFNβ treatment (Fig. 2C). Under these conditions, transcription of *Id1* and *Id3*, two canonical SMAD-responsive genes, was induced in HFF-1 upon BMP and Activin B stimulation, but not after IFNβ treatment (Fig. 2D). This confirms that HFF-1 express functional BMP type I and type II receptor signaling pathways.

Next, we analyzed whether BMP signaling in HFF-1 may induce other target genes than *Id1* and *Id3*, and analyzed the transcript level of interferon-stimulated genes (ISGs) upon BMP stimulation, which was demonstrated previously in other cell types (Eddowes et al, 2019). Importantly, the ISGs *Isg15*, *Irf9*, *Ifi6*, and *Stat2* were upregulated upon stimulation with BMPs and Activins, albeit mostly to a lower level compared to IFNβ stimulation (Fig. 2E). These data verify that BMP- and Activin-mediated signaling in HFF-1 is biologically active and leads to transcriptional activation of canonical BMP targets as well as ISGs.

## BMP9 enhances the antiviral response of HFF-1

Since stimulation of HFF-1 with BMPs or Activin B induced transcription of ISGs, we assessed whether this results in antiviral activity against HCMV. For this, we either stimulated HFF-1 with BMPs or Activin B or co-stimulated with BMPs/Activin B and IFNβ for 6 h, and then infected them with HCMV (Fig. 3A). Analysis of immediate-early 1 (IE1) protein expression, which is a marker for early HCMV gene expression, revealed that stimulation with BMP15 or Activin B significantly reduced the number of HCMV IE1+ cells, independently of added IFNβ (Fig. 3B). BMP4 and BMP6 did not affect the number of HCMV IE1+ cells alone (Fig. 3B, left panel) or in combination with IFNβ (Fig. 3B, right panel). Interestingly, while stimulation with BMP9 alone did not affect the number of HCMV IE1+ cells (Fig. 3B, left panel), co-stimulation of HFF-1 with BMP9 and IFNβ led to significantly lower IE1+ counts compared to IFNβ stimulation alone (Fig. 3B, right panel). This effect was maintained at high and low concentrations of BMP9 (Fig. 3C). Due to this dependence of the antiviral effect of BMP9 on IFNβ, we then focused our study on the investigation of this phenotype, rather than on the BMP15/Activin B phenotype, which we will investigate in the future.

With the potential involvement of BMP9 in the innate immune response, we next investigated whether BMP9 is secreted in response to HCMV infection. For this, we infected HFF-1 with HCMV and collected supernatants in 6 hour increments. To monitor the presence of BMPs in the supernatant of HCMV-

infected HFF-1, 293T, which also express all tested BMP receptors (Fig. EV1A, B), were transfected with expression plasmids for a BRE-Luciferase reporter and treated with the supernatants from HCMV-infected HFF-1. The BRE-luciferase reporter responds to BMP stimulation with transcriptional activation of the luciferase gene (Fig. 3D). In addition, supernatants were incubated with a specific neutralizing antibody against BMP9 (Fig. EV1C, D) prior to stimulation of 293T to identify which portion of the induced response can be attributed to BMP9. As shown in Fig. 3D, upon treatment of 293T with HFF-1 supernatant harvested at early and late stages of HCMV infection (0–12 and 42–54 h post infection (hpi)), BRE-Luciferase fold induction was higher than in cells that were treated with supernatant from mock-infected HFF-1. Notably, the majority of the BRE response induced by the supernatants could be neutralized by the addition of the BMP9 antibody (Fig. 3D, dark gray symbols), suggesting that infection of HFF-1 with HCMV induces secretion of BMP9, which may in turn affect the innate immune response to HCMV infection.

## BMP9 stimulation modulates the transcription of critical components of the IFNAR signaling pathway

With BMP9 enhancing the antiviral activity of HFF-1 against HCMV infection when co-applied with IFNβ (Fig. 3B,C), we next evaluated whether BMP9 stimulation enhances IFNβ-mediated signaling, or vice versa (Fig. 4A). For this, we analyzed transcript levels of the BMP9-associated receptors ALK1 (encoded by *Acvrl1*) and BMPR2 (encoded by *Bmpr2*) (Jiang et al, 2021; Salmon et al, 2020), as well as the SMAD-responsive gene *Id1*, in cells stimulated with BMP9 alone or in combination with IFNβ. No significant change in expression was detected when both stimuli were added (Fig. 4B), indicating that IFNβ stimulation did not enhance BMP9-mediated signaling. Notably, while expression levels of some ISGs, namely *Isg15*, *Irf7*, and *Ifi6*, were not affected upon co-stimulation with BMP9 and IFNβ compared to IFNβ stimulation alone (Fig. 4C, upper panels), transcript levels of the ISGs *Irf9*, *Stat2*, and *Irf1* were significantly higher in the co-stimulation setting (Fig. 4C, lower panels). Congruently, expression of genes encoding negative regulators of BMP signaling (*Smurf1* (Zhu et al, 1999)) or IFNAR signaling (*Usp18* and *Smurf1*, (Arimoto et al, 2017; Yuan et al, 2012) were significantly lower in co-stimulated HFF-1 than in cells stimulated with IFNβ alone (Fig. 4D). This indicates that BMP9 modulates transcription of genes related to the antiviral immune response, especially transcripts associated with IFNAR signaling. We next applied inhibitors of canonical BMP- or IFNAR-mediated signaling, DMH1 and Ruxolitinib, respectively, to assess whether *Stat2* and *Irf9* transcription depended on BMP9-mediated signaling

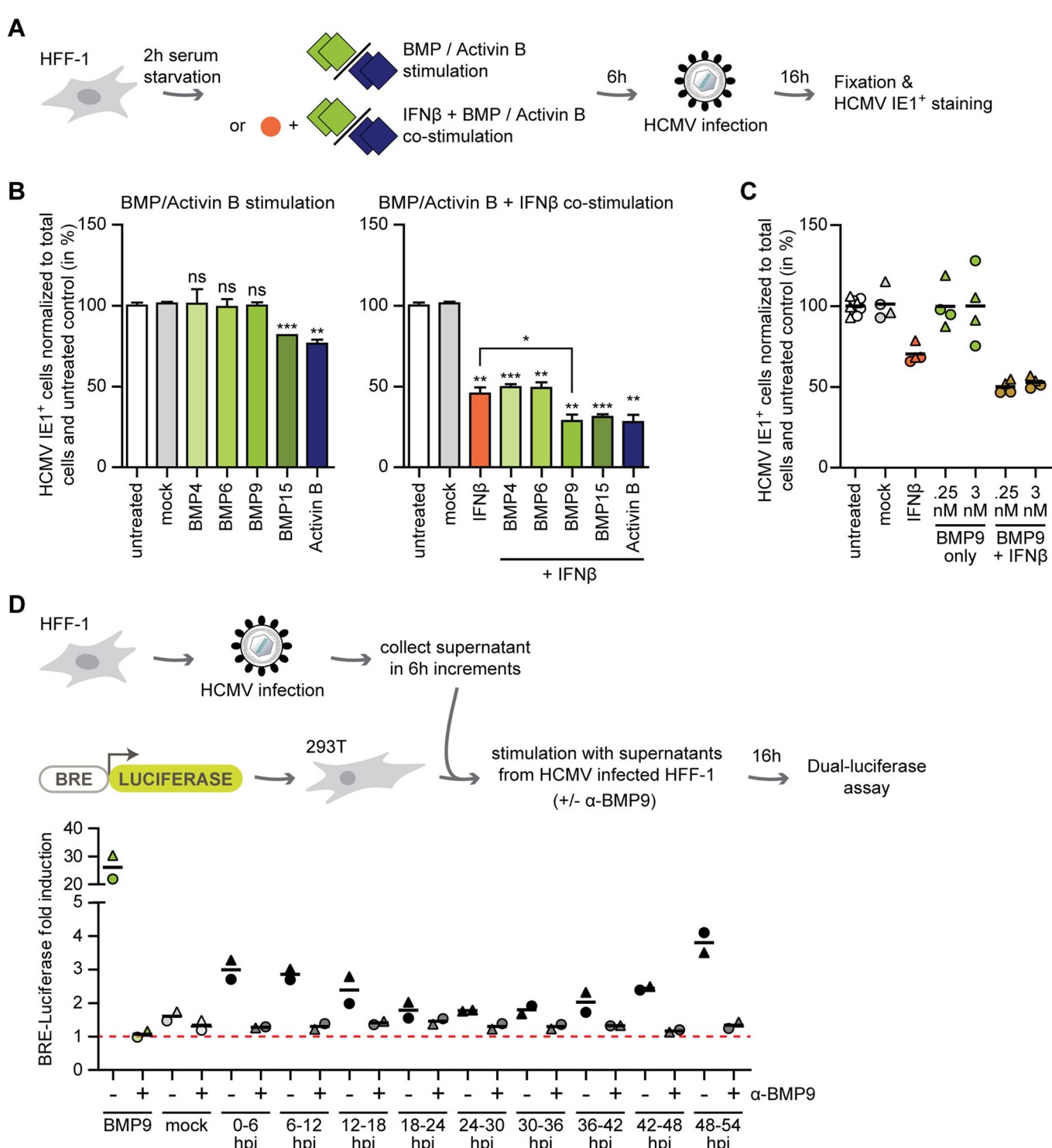

(Fig. 4E). While DMH1, but not Ruxolitinib, negated BMP9-mediated induction of *Id3* (Figs. 4E and EV1E), transcription of *Stat2* and *Irf9* upon BMP9 stimulation was independent of canonical BMP signaling, but dependent on intact IFNAR signaling. The same result was obtained in the BMP9/IFNβ co-stimulation context (Fig. 4E). Collectively, these data suggest that BMP9-induced enhancement of ISG expression is mediated by IFNAR signaling.

## BMP9 stimulation enhances activation of STAT1 upon type I IFN, but not type II IFN stimulation

Phosphorylation of the transcription factors STAT1 and STAT2 is a hallmark for the activation of IFNAR signaling. As Ruxolitinib treatment prevented BMP9-mediated induction of ISG transcription (Fig. 4E), we hypothesized that BMP9-mediated signaling may directly affect IFNAR signaling. To investigate this, we stimulated

◀ **Figure 3. BMP9 enhances the antiviral response of HFF-1 to HCMV infection.**

(A) Schematic representation of the workflow. After 2 h of serum starvation, HFF-1 were stimulated for 6 h with BMP4 (18 nM), BMP6 (18 nM), BMP9 (3 nM), BMP15 (18 nM), or Activin B (4 nM), or co-stimulated with IFNβ (5 ng/ml), followed by HCMV infection (MOI 0.5) for 16 h. Cells were fixed, nuclei were stained and cells were labeled for HCMV IE1+ cells as a readout for infection. (B) HCMV IE1+ cells normalized to total cell numbers and the untreated control (white column) in BMP/Activin stimulated samples (left panel) or with IFNβ co-stimulated samples (right panel). (C) HCMV IE1+ cells normalized to total cell numbers in cells pre-stimulated with either low (0.25 nM) or high (3 nM) concentrations of BMP9 (green symbols) or IFNβ co-stimulated with low and high concentrations of BMP9 (beige symbols). (D) HFF-1 were infected by centrifugal enhancement with HCMV WT (MOI 0.5) and supernatants of infected cells were collected in 6 h increments. 293T were co-transfected with expression plasmids for the BRE-Luciferase reporter and a Renilla luciferase normalization control. Twenty-four hours post transfection, 293T were either stimulated with supernatants from HCMV-infected cells, or supernatants from HCMV-infected cells incubated for 15 min at RT with an α-BMP9 antibody, for 16 h, followed by a dual-luciferase assay readout. Luciferase fold induction was calculated by dividing Renilla-normalized values from stimulated samples by the corresponding values from unstimulated samples. Data information: (B) Experiment was performed three independent times, one representative is shown. (C, D) Data are combined from two independent experiments. Student's t test (unpaired, two-tailed), n.s. not significant, *$P < 0.05$, **$P < 0.01$, ***$P < 0.001$. Data are shown as mean ± SD. Source data are available online for this figure.

HFF-1 with either BMP9, IFNβ or in combination with decreasing concentrations of IFNβ and analyzed STAT1 phosphorylation 1 and 8 h later (Fig. 5A). While BMP9 alone did not lead to detectable levels of STAT1 phosphorylation, co-stimulated HFF-1 displayed stronger phosphorylation of STAT1 compared to HFF-1 stimulated with IFNβ alone at both time points (Fig. 5A, quantification in graph). Substituting IFNβ with IFNα, another type I IFN, led to comparable results, while BMP9 did not affect responses mediated by IFNγ, a type II IFN (Fig. 5B). Congruently, transcription of *Irf9*, *Stat2*, and *Irf1* was enhanced upon addition of BMP9 to IFNα or IFNβ stimulated cells, but not when applied with IFNγ (Fig. 5C). Transcription of the SMAD-responsive gene *Id3* was not affected by addition of any of the IFNs (Fig. 5C). These results reveal that BMP9 modulates type I IFN-, but not type II IFN-mediated signaling, likely by increasing STAT1 phosphorylation.

## HCMV US18 and US20 synergistically inhibit BMP-mediated signaling

HCMV US18 and US20 are transmembrane proteins harboring seven transmembrane domains (Fig. EV2A), and have been associated, via a proteomic screen, with downregulation of the type I BMP receptors ALK1 and ALK2 during HCMV infection (Fielding et al, 2017). This is of particular interest as ALK1 is one of the main type I BMP receptors associated with BMP9-mediated signaling (Salmon et al, 2020). Thus, we investigated whether the expression of HCMV US18 and US20 specifically inhibits BMP9-mediated signaling without affecting type I IFN signaling pathways. For this, we transfected 293T with the reporter plasmids BRE-Luciferase or MX1-Luciferase, responsive to BMP or IFNβ stimulation, respectively, and stimulated them with BMP9, IFNβ or both (Fig. 6A). To verify our reporters, we included the M27 protein of murine CMV, a well-characterized inhibitor of IFNAR signaling (Zimmermann et al, 2005), and expression of all proteins of interest was verified (Fig. 6C). Notably, expression of HCMV US18 or US20 inhibited BMP9-mediated induction of the BRE-Luciferase reporter, but not of the MX1-Luciferase reporter (Fig. 6B), and co-expression of US18 and US20 inhibited BMP9 signaling even stronger (Fig. 6D). Moreover, we could exclude modulatory effects of US18 and US20 expression on pattern recognition receptor (PRR) signaling (cGAS-STING, RIG-I or downstream of the PRR-activated transcription factor IRF3) (Fig. 6E), suggesting specific downregulation of BMP9-induced, but not of PRR- or IFNAR-induced signaling.

To verify these observations in HFF-1, we generated HFF-1 with doxycycline-inducible expression of V5-tagged US18 and/or HA-tagged US20 (Fig. 7A). Expression of HCMV US20 in HFF-1

showed a double-band likely corresponding to N-linked glycosylation since PNGase F treatment resulted in a single band (Fig. 7B). We next assessed whether expression of US18-V5 and/or US20-HA in HFF-1 can blunt the response to BMP9 stimulation. Indeed, in the presence of US18/US20, phosphorylation of SMAD1/5/9 was not detectable (Fig. 7C), and *Id3* transcription was significantly reduced after BMP9 stimulation (Fig. 7D, upper graph), in line with the observation that both downregulate cell surface expression of BMPRs (Fielding et al, 2017). Strikingly, expression of either US18-V5 or US20-HA was sufficient to abrogate BMP9-mediated enhancement of *Stat2* transcription, while they did not affect IFNβ-induced activation of *Stat2* transcription (Fig. 7D, lower graph). These data show that expression of HCMV US18 and US20 in HFF-1 specifically inhibits BMP9-mediated signaling, as well as BMP9-mediated enhancement of IFNAR signaling, while they do not affect IFNAR signaling.

## US18 and US20 are components of the HCMV virion

Next, we evaluated the role of US18 and US20 during HCMV infection. For this, we generated recombinant viruses with either V5-tagged US18 or HA-tagged US20 since antibodies against US18 and US20 are not available. Interestingly, US18-V5 and US20-HA proteins were detected in viral particles by immunoblotting (Fig. EV2B), indicating that US18 and US20 are components of the virions. Moreover, US18-V5 and US20-HA were already detected 3 h post infection (hpi) in HFF-1 (Fig. EV2C), too early for de novo expressed viral proteins, suggesting that US18 and US20 were delivered into cells. De novo expression of US18-V5 and US20-HA was detectable from 24 hpi (Fig. EV2D), which is in line with previous studies (Cavaletto et al, 2015; Fielding et al, 2014). Interestingly, the expression of US18 and US20 during HCMV infection overlapped with the time windows when we detected BMP9 secretion of HCMV-infected cells (Fig. 3D), indicating a concurrence of the host-generated BMP9-mediated enhancement of the antiviral immune response with the antagonistic virus-encoded inhibitors of BMP signaling.

## HCMV US18 and US20 inhibit BMP9-mediated signaling during infection

Next, we evaluated the impact of US18 and US20 expression on BMP9-induced signaling during HCMV infection. For this, we generated HCMV recombinants lacking US18, US20 or both proteins by introducing stop codons in the respective genomic

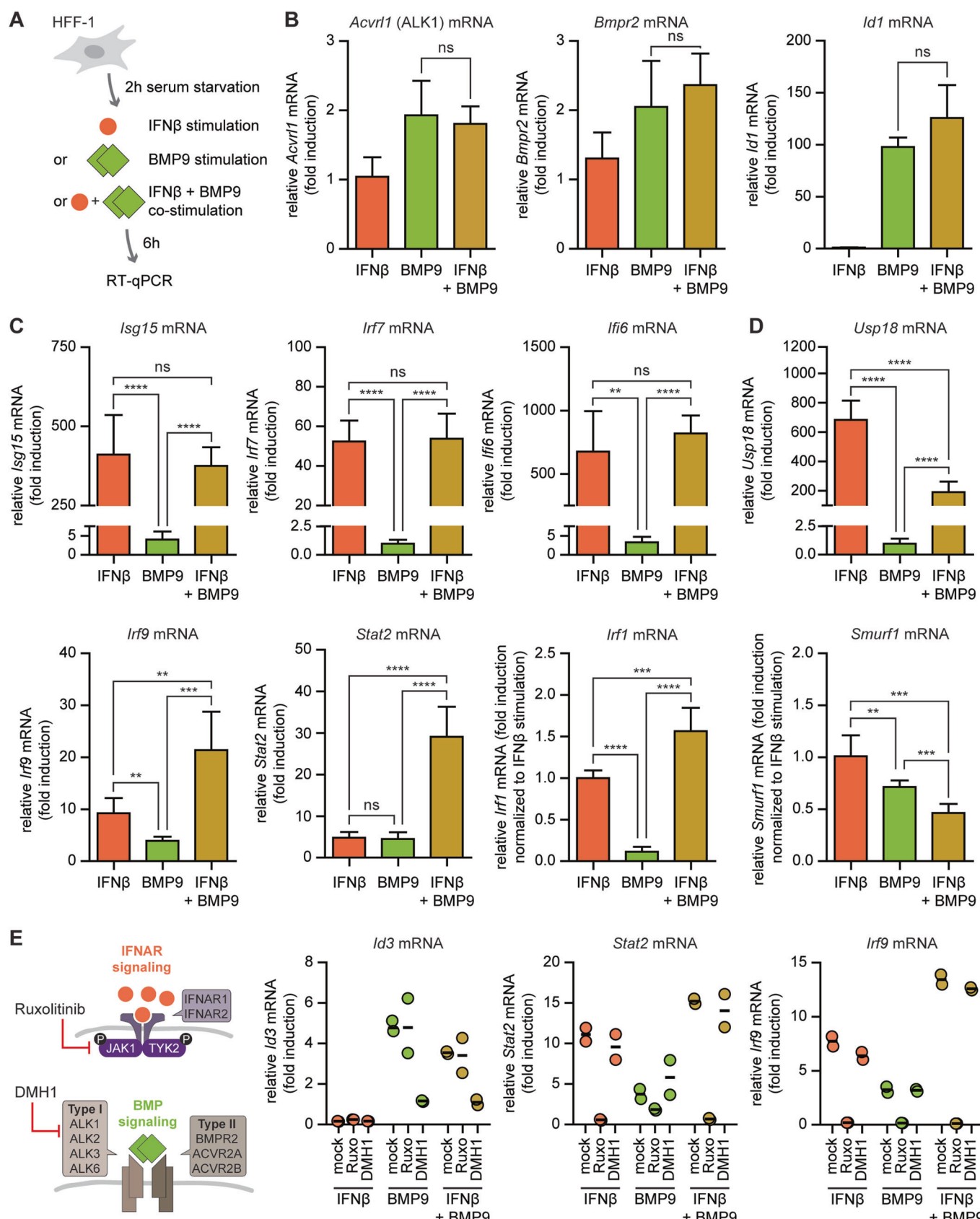

**Figure 4.  BMP9 modulates transcription of critical components of the IFNAR signaling pathway.**

(A) Schematic representation of the workflow. After 2 h of serum starvation, HFF-1 were stimulated for 6 h with either IFNβ (5 ng/ml) or BMP9 (3 nM) alone, or co-stimulated with IFNβ and BMP9, followed by RNA extraction from cell lysates and RT-qPCR. (B) Transcript levels of BMP receptors *Acvrl1* (ALK1, type I receptor) and *Bmpr2* (type II receptor), which are the main receptors for BMP9, and the BMP-responsive gene *Id1*. (C) Transcript levels of the ISGs *Isg15, Irf7, Ifi6, Irf9, Stat2,* and *Irf1*. (D) Transcript levels of the negative regulators *Usp18* and *Smurf1*. (E) HFF-1 were incubated with either Ruxolitinib or DMH1 for 1 h, then stimulated for 6 h with either IFNβ (5 ng/ml) or BMP9 (3 nM) alone, or co-stimulated with IFNβ and BMP9, followed by RNA extraction from cell lysates and RT-qPCR for transcript levels of *Id3, Stat2,* and *Irf9*. Data information: (B–D) Data are combined from three independent experiments with the exception of *Ifi6* transcript levels where three independent experiments were performed and data was combined from two independent experiments. (E) The experiment was performed two independent times, one representative is shown. Student's *t* test (unpaired, two-tailed), n.s. not significant, **$P < 0.01$, ***$P < 0.001$, ****$P < 0.0001$. Data are shown as mean ± SD. Source data are available online for this figure.

locus, and designated them as HCMV US18stop, US20stop, and US18/20stop. Since US18, US19, and US20 are transcribed as a tri-cistronic mRNA (Guo and Huang, 1993), this approach allowed minimal editing of the viral genome and ensured that only the proteins of interest were not expressed. With these viruses, we infected HFF-1 with a high or low multiplicity of infection for 3 h or 48 h, respectively, then stimulated the infected cells with BMP9 or mock-treated them for either 2 h for immunoblotting or 6 h for RT-qPCR analysis (Fig. 8A). At early stages of infection (3 hpi), HFF-1 infected with either HCMV US18stop, HCMV US20stop, or HCMV US18/20stop displayed a strong phosphorylation of SMAD1/5/9 already in the absence of BMP9 stimulation, potentially due to endogenously secreted BMP9 (see Fig. 3D). Notably, this was not visible in HCMV WT-infected cells (Fig. 8B). This suggests that incoming US18 and US20 (likely present in the virion, see Fig. EV2B, C) are involved in downregulation of BMP signaling. Interestingly, US18/US20-mediated inhibition of BMP signaling could be overcome in the presence of an excess of exogenous BMP9, visible by detectable levels of SMAD1/5/9 phosphorylation in HCMV WT-infected cells (Fig. 8B). In line with these results, HFF-1 infected with HCMV WT induced *Id3* and *Stat2* transcription in response to BMP9 stimulation, while HFF-1 infected with HCMV US18stop, HCMV US20stop, or HCMV US18/20stop induced transcription of *Id3* and *Stat2* independently of BMP9 (Fig. 8B).

At late stages of infection (48 hpi), when US18 and US20 are de novo expressed (see Fig. EV2D), BMP9 stimulation led to higher SMAD1/5/9 phosphorylation levels in HFF-1 infected with HCMV lacking US18/20 compared to HCMV WT, indicating that also de novo expressed US18/US20 downregulate BMP signaling at this time point of infection (Fig. 8C). Congruently, HCMV WT-infected HFF-1 did not respond to BMP9 stimulation with changes in *Id3* and *Stat2* transcript levels, while HFF-1 infected with HCMV US18stop, HCMV US20stop, or HCMV US18/20stop readily responded to BMP9 stimulation with an increase in *Id3* and *Stat2* transcripts (Fig. 8C).

In summary, our data shows for the first time that US18 and US20 downmodulate BMP9-mediated signaling at early and late stages of HCMV infection.

## US18 and US20 dampen the antiviral immune response to HCMV infection

Since BMP9 enhances the antiviral immune response and HCMV US18/US20 downmodulate BMP9-mediated signaling, we raised the hypothesis that HCMV lacking US18 and US20 expression could be more sensitive to IFNβ treatment than WT HCMV. To

test this, we infected HFF-1 with either HCMV WT or HCMV US18/20stop for 3 h, followed by stimulation with either BMP9, low-dose IFNβ, or co-stimulation with BMP9 and low-dose IFNβ (Fig. 9A). In line with Fig. 8B, HFF-1 infected with HCMV US18/20stop induced transcription of *Id3* independent of BMP9, and correlating with data from Figs. 4 and 7, stimulation with IFNβ did not affect *Id3* transcript levels in any condition tested (Fig. 9B). Transcript levels of the ISGs *Irf9* and *Stat2* in HFF-1 infected with HCMV WT were significantly higher in the co-stimulation context compared to IFNβ stimulation alone (Fig. 9C), resembling the phenotype observed in Fig. 4. Strikingly, this enhanced transcription of *Irf9* and *Stat2* was associated with lower transcript levels of the viral *IE1* and *UL44* transcripts in HCMV WT-infected cells (Fig. 9D). In comparison, IFNβ stimulation of HFF-1 infected with HCMV US18/20stop induced higher transcript levels of *Irf9* and *Stat2* compared to HCMV WT-infected cells (Fig. 9C). This indicates that the lack of US18 and US20 at early stages of HCMV infection, which in turn leads to activated BMP-mediated signaling (Fig. 8B), promotes stronger immune signaling in response to IFNβ stimulation. Congruently, HFF-1 infected with HCMV US18/20stop displayed significantly lower transcript levels of viral *IE1* and *UL44* upon IFNβ stimulation compared to HFF-1 infected with HCMV WT (Fig. 9D). To rule out that the differences in viral transcript levels are caused by different infection efficiencies of WT and US18/20stop viruses, we compared the expression level of the tegument protein UL82/pp71, which is a component of the viral particle and delivered into infected cells, 3 h post infection between the two viruses. As shown in Fig. EV3, UL82 expression was comparable among HCMV WT and HCMV US18/20stop.

In summary, our data show that an HCMV mutant lacking US18 and US20 expression is more sensitive to IFNβ treatment at early stages of infection, consistent with an important involvement of BMP9 in the antiviral immune response to HCMV infection.

## Discussion

Through millions of years of co-evolution, herpesviruses have developed effective strategies to moderate immune control for securing lifelong persistence in their respective hosts. The diverse mechanisms by which HCMV evades innate immune control are still incompletely understood, and bear the potential to discover novel and yet unanticipated roles of cellular pathways and their involvement in HCMV pathogenesis. The presence of potential viral BMP signaling modulators (Fielding et al, 2017) and a new role for BMPs in cellular antiviral immunity (Eddowes et al, 2019;

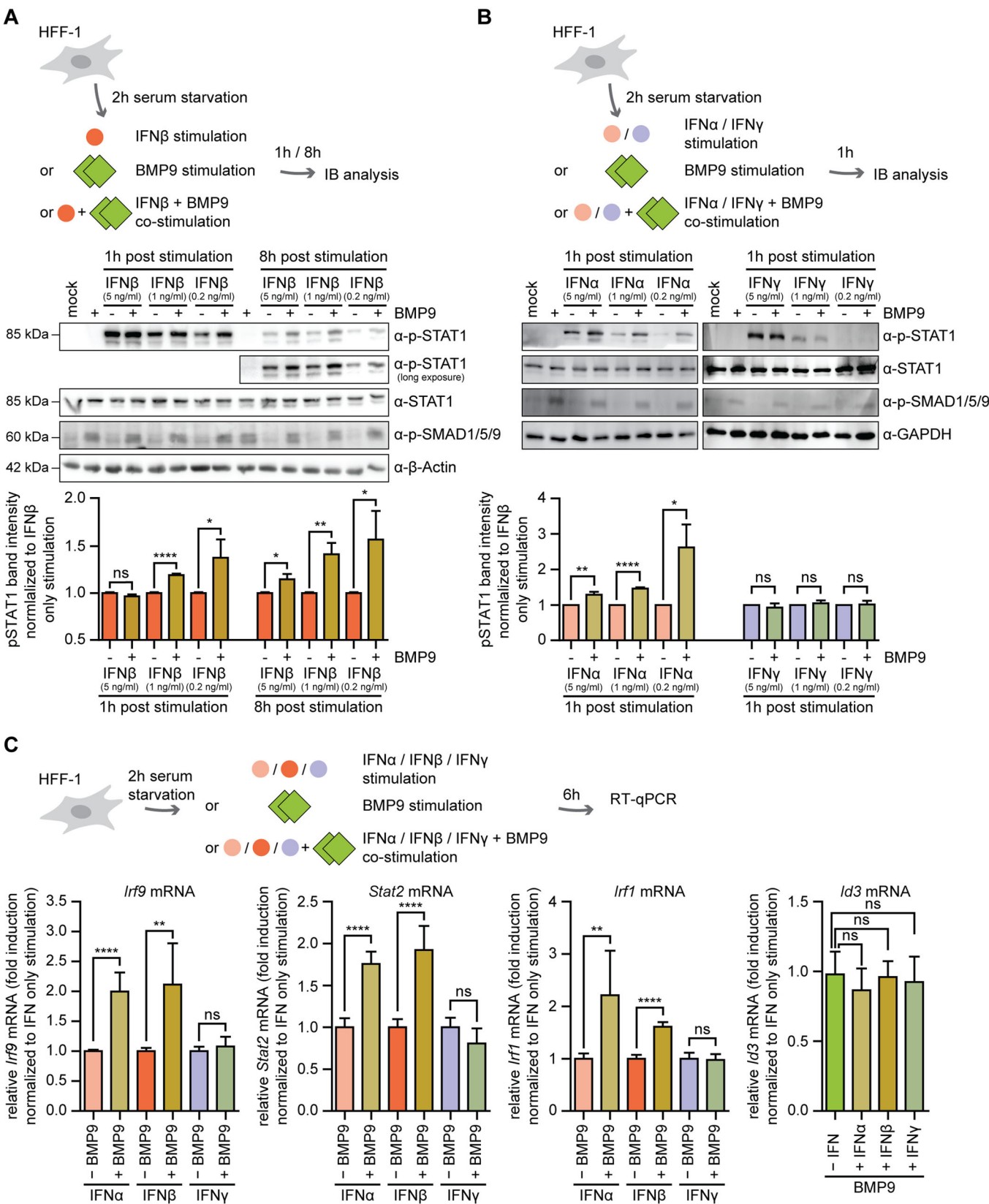

**◄ Figure 5.    BMP9 enhances STAT1 activation upon type I IFN, but not type II IFN stimulation.**

(A) After 2 h of serum starvation, HFF-1 were stimulated for either one or 8 h with IFNβ (5 ng/ml) or BMP9 (3 nM) alone, or were co-stimulated with IFNβ and BMP9, followed by cell lysis and immunoblot analysis with antibodies for phospho-STAT1, STAT1, phospho-SMAD1/5/9, and β-Actin. Phospho-STAT1 band intensities were first normalized to corresponding total STAT1 levels, then to IFNβ stimulation only, and are shown below. (B) After 2 h of serum starvation, HFF-1 were stimulated for 1 h with either IFNα2 (5 ng/ml), IFNγ (5 ng/ml), or BMP9 (3 nM) alone, or co-stimulated with IFNα2 or IFNγ and BMP9, followed by cell lysis, immunoblot analysis and quantification as in (A). (C) After 2 h of serum starvation, HFF-1 were stimulated for 6 h with either IFNα2 (5 ng/ml), IFNβ (5 ng/ml), IFNγ (5 ng/ml), or BMP9 (3 nM) alone, or co-stimulated with IFNα2, IFNβ, IFNγ, and BMP9, followed by RNA extraction from cell lysates and RT-qPCR for *Irf9*, *Stat2*, *Irf1*, and *Id3* transcripts. Data information: (A, B) Experiment was performed three independent times, one representative immunoblot is shown. Quantified data for the STAT1 phosphorylation levels are combined from three independent experiments. (C) Data are combined from three independent experiments. Student's *t* test (unpaired, two-tailed), n.s. not significant, \*P < 0.05, \*\*P < 0.01, \*\*\*\*P < 0.0001. Data are shown as mean ± SD. Source data are available online for this figure.

Jiyarom et al, 2022; Poole et al, 2021; Zhong et al, 2021) stimulated us to investigate BMP-HCMV interactions in detail.

We found that several BMPs stimulated the transcription of selected ISGs, and discovered that in combination with IFNβ, BMP9 in particular enhanced restriction of HCMV replication and boosted the transcriptional response to IFNβ. We also show for the first time that BMP9 was secreted by human fibroblasts upon HCMV infection. Notably, BMP9 is the only BMP that has been confirmed to circulate at active concentrations in the host (Bidart et al, 2012; Herrera and Inman, 2009) and the majority of secreted BMP9 is produced in the liver, a target organ of HCMV (Bidart et al, 2012; Griffiths and Reeves, 2021). Interestingly, we also observed that BMP15 and Activin B had effects on HCMV, and this will be pursued in future work.

We found that HCMV US18 and US20 impaired the BMP9-induced enhancement of the response to IFNβ, consistent with their reported activity downregulating the type I BMPRs ALK1 and ALK2 (Fielding et al, 2017). Interestingly, BMP9 is one of only two BMPs, with BMP10 being the second one, which activates the ALK1/2-BMPR2 signaling cascade (David et al, 2007). BMPs bind to type I receptors first and then recruit a type II receptor. Hence, by inactivating ALK1 and ALK2, HCMV can prevent BMP9-mediated signaling, but could potentially leave BMPR2-mediated signaling still able to operate. Recently, BMPR2 was reported to be crucial for the establishment of latency (Poole et al, 2021). Therefore, downregulation of ALK1 and ALK2 by US18 /US20 may impair the antiviral effects of BMP9, but allow signaling by other BMP/Activin/TGF-β ligands (Yu et al, 2005), which may preserve the ability of HCMV to establish latent infections. It will be important in the future to investigate the relationship between HCMV US18/20 and BMP signaling during HCMV latency and reactivation.

We identified cross-talk between BMP9-induced and IFNAR-mediated signaling. Others have reported similar effects in different contexts, although the underlying molecular mechanisms may differ. In Eddowes et al, BMP6-induced SMAD complexes were proposed to occupy SMAD-binding elements in the promotor region of ISGs, thus acting independently of IFN/IFNAR signaling (Eddowes et al, 2019); however, downregulation of the IFNAR signaling inhibitor USP18 by BMP6 was also observed, as in our case for the BMP9/IFNβ co-stimulation context. In our experiments, BMP9-mediated induction of ISGs was completely abrogated when using the Jak1 inhibitor Ruxolitinib, but not with DMH1 which inhibits canonical SMAD signaling. This indicates that BMP9 may activate the IFNAR signaling pathway somewhere at the kinase level. However, our analyses did not find a detectable level of phosphorylated STAT1 upon BMP9 stimulation, so either the BMP9-induced ISG transcription is independent of STAT1, or the amount of phosphorylated STAT1 is

below the limit of detection by immunoblot. We observed down-regulation of Smurf1 upon BMP9/IFNβ co-stimulation, in line with a previous study (Sreekumar et al, 2017). This effect could act as booster for IFNAR signaling since Smurf1 is also a negative regulator of IFNAR signaling (Yuan et al, 2012). The differential transcription upon BMP9 only versus BMP9/IFNβ co-stimulation (some ISGs not affected, other ISGs upregulated, negative IFNAR regulators down-regulated) suggests that a specific regulation of transcriptional activation upon stimulation is present. Others have also investigated roles of BMPs in innate antiviral immune signaling in different contexts. In zebrafish, BMP8a was found to be essential for defence against RNA virus infection via promoting phosphorylation of the PRR signaling pathway components TBK1 and IRF3 through the p38 MAPK pathway, thereby activating the type I IFN response (Zhong et al, 2021). Based on our results with the BMPR antagonists US18 and US20, we think that PRR-mediated induction of type I IFN transcription is unlikely to be involved during BMP9 signaling. A recent study on human Sertoli cells showed that Zika virus infection can counteract BMP6 signaling, and that BMP6 induces phosphoryla-tion of IRF3 and STAT1, and increases expression of IFNβ and some ISGs (Jiyarom et al, 2022). In general, these studies all suggest that interactions between BMP and type I IFN pathways can occur at multiple juncture points that may vary depending on organism and cell type and represent a key modulator of antiviral immune responses that is targeted by viruses.

In summary, we find the first evidence of an important interaction between BMP9, HCMV infection, and type I IFN-mediated antiviral pathways. BMP9 secretion is induced by HCMV infection, and enhances the transcriptional response to and antiviral activity of IFNβ, but its activity is counteracted by HCMV-encoded proteins US18 and US20 that downregulate type I BMP receptors. These results, along with data derived from studying BMPs in the context of other viruses, suggest that the BMP pathway is an underappreciated modulator of innate immunity to viral infection. The BMP pathway is pleiotropic with multiple roles across physiology at all life stages, raising the possibility that viral inhibition of BMP signaling not only regulates the innate immune response and viral replication, but could also have effects on development, tissue repair and homeostatic mechanisms.

# Methods

## Cell lines

The primary human foreskin fibroblast (HFF-1; SCRC-1041), MRC-5 (CCL-171) and human embryonic kidney 293T (293T; CRL-3216) cell lines were obtained from ATCC. 293T and MRC-5

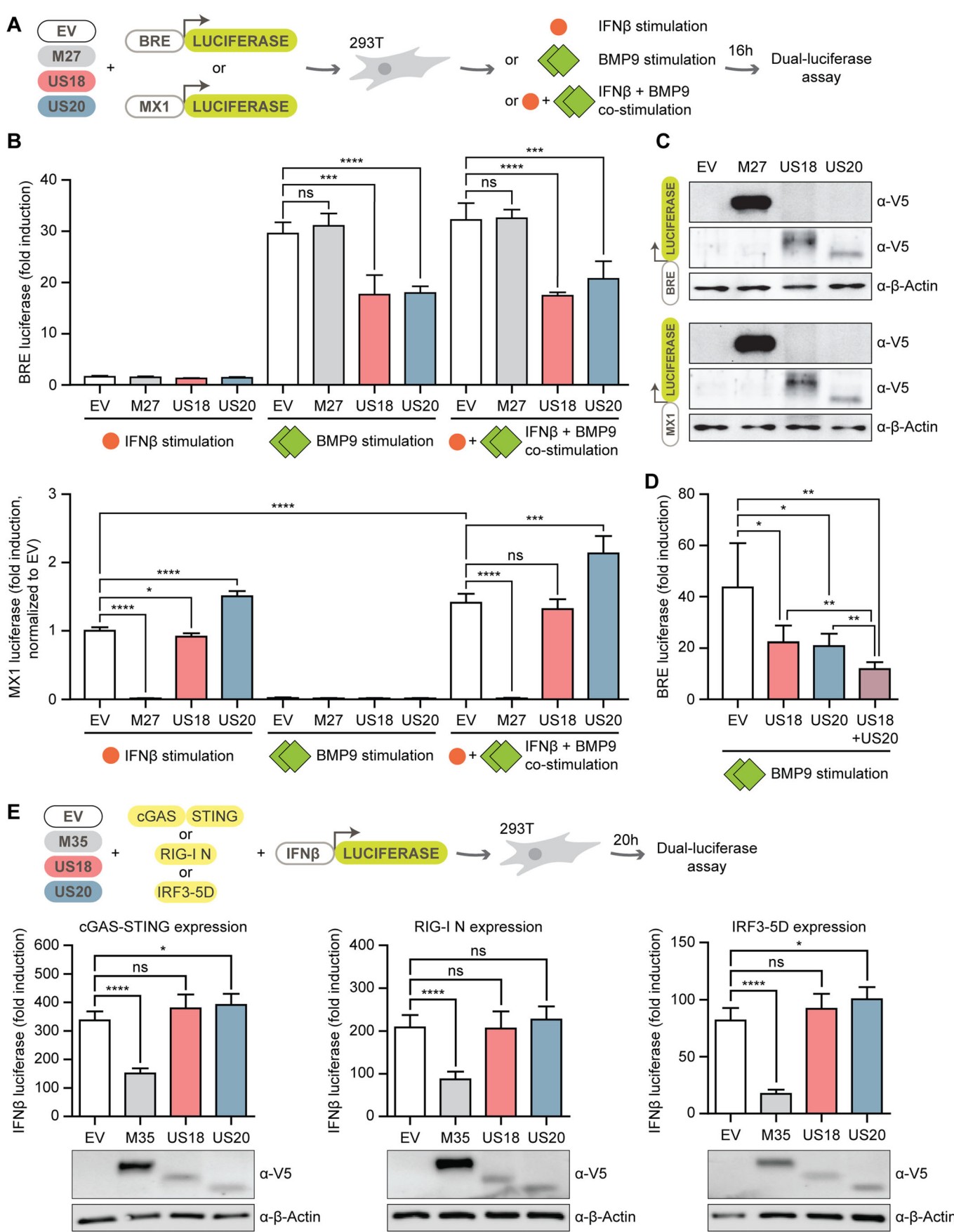

**Figure 6. HCMV US18 and US20 specifically inhibit BMP9-mediated, but not PRR- or IFN-mediated signaling.**

(A) 293T were co-transfected with expression plasmids for empty vector (EV) or V5-tagged US18, US20, or M27 (a known inhibitor of IFNAR signaling), together with either a BRE-Luciferase or MX1-Luciferase reporter and a Renilla luciferase normalization control. Twenty-four hours post transfection, 293T were stimulated for 16 h with either IFNβ (5 ng/ml) or BMP9 (3 nM) alone, co-stimulated with IFNβ and BMP9, or left unstimulated, followed by cell lysis and a Dual-luciferase assay readout. (B) Results from the Dual-luciferase assay with the BRE-Luciferase (top panel) and MX1-Luciferase (bottom panel) reporter. (C) Cell lysates from (B) were analyzed by immunoblot for the expression of M27, US18 and US20 with a V5-specific antibody, β-Actin served as loading control. (D) 293T were co-transfected with expression plasmids for the BRE-Luciferase and Renilla reporters as in (A), together with V5-tagged US18, US20 or co-transfected with US18 and US20 in combination. 24 h post transfection 293 T were stimulated for 16 h with BMP9 (3 nM), or left unstimulated, followed by cell lysis and a Dual-luciferase assay readout. (E) 293T were co-transfected with expression plasmids for either Cherry-STING and cGAS-GFP (left panel), RIG-I N (middle panel), or IRF3-5D (a constitutively activate IRF3 mutant; right panel), together with the murine IFNβ-luciferase reporter (IFNβ-Luc) and the Renilla reporter as normalization control. Cells were additionally transfected with expression plasmids for EV, V5-tagged M35 (a known inhibitor of PRR-mediated signaling pathways, (Chan et al, 2017), US18 or US20. Twenty hours post-transfection, cells were lysed and a dual-luciferase assay was performed. Immunoblot analysis of cell lysates from the respective experiments for M35, US18, and US20 detected with a V5-specific antibody, and β-Actin as a loading control, are shown below. Data information: (B–E) Data are combined from three independent experiments, for the immunoblots one representative is shown. Student's *t* test (unpaired, two-tailed), n.s. not significant, *$P < 0.05$, **$P < 0.01$, ***$P < 0.001$, ****$P < 0.0001$. Data are shown as mean ± SD. Luciferase fold induction was calculated by dividing Renilla-normalized values from stimulated samples by the corresponding values from unstimulated samples. Source data are available online for this figure.

were maintained in Dulbecco's modified Eagle's medium (DMEM; high glucose) supplemented with 8% fetal calf serum (FCS) and 1% penicillin/streptomycin (P/S). HFF-1 were maintained in DMEM (high glucose) supplemented with 15% FCS, 1% P/S, and 1% non-essential amino acids (NEAA). Cells were cultured at 37 °C and 7.5% $CO_2$. To generate HFF-1 with doxycycline-inducible expression of US18-V5 and/or US20-HA, lentiviral transduction was performed. For this, 293T (730,000 cells per well, six-well format) were transfected with 600 ng psPAX2, 600 ng pCMV-VSV-G and 800 ng of either pW-TH3 empty vector (EV1), pW-YC1 empty vector (EV2), pW-TH3 US18-V5, or pW-YC1 US20-HA complexed with Lipofectamine 2000. Sixteen hours post transfection, medium was changed to lentivirus harvest medium (DMEM h.gl. supplemented with 20% FCS, 1% P/S, and 10 mM HEPES). Forty-eight hours post transfection, lentivirus was harvested, diluted 1:2 with HFF-1 medium, and polybrene was added to a final concentration of 4 µg/ml. HFF-1 were seeded the day before transduction in a six-well format with 250,000 cells per well. For transduction, HFF-1 medium was replaced by medium containing lentivirus, and cells were infected by centrifugal enhancement at 684 × *g* and 30 °C for 90 min. Three hours post infection, medium was replenished with fresh HFF-1 medium. Successfully transduced cells were selected with 2 µg/ml Puromycin (HFF-1 transduced with pW-TH3 vectors) or 150 µg/ml Hygromycin (HFF-1 transduced with pW-YC1 vectors). Selection of cells with puromycin lasted over ~1.5 weeks with at least three passages, and selection with hygromycin over ~2.5 weeks with at least five passages. Untransduced control cells were treated with the antibiotics side-by-side to evaluate when the selection reached its endpoint. Occasionally, antibiotics were added to the medium when cells were passaged. Experiments were always performed with cells that were not treated with antibiotics for at least one passage prior the experiment.

## Viruses

The wild-type HCMV TB40-BAC4 (hereinafter designated as HCMV WT) was characterized previously (Sinzger et al, 2008) and kindly provided by Martin Messerle (Institute of Virology, Hannover Medical School, Germany). HCMV BACs were reconstituted after transfection of MRC-5 with purified BAC DNA. The reconstituted virus was propagated in HFF-1 and virus was purified on a 10% Nycodenz cushion. The resulting virus pellets were

resuspended in virus standard buffer (50 mM Tris–HCl pH 7.8, 12 mM KCl, 5 mM EDTA) and stored at −70 °C. Infectious titer was determined by standard plaque assay and IE1 labeling in HFF-1. Manipulation of the HCMV genome was carried out by *en passant* mutagenesis (Tischer et al, 2010). pEP-KanS (Tischer et al, 2010) served as the template for PCR. For construction of the recombinant HCMV US18stop, a linear PCR product was generated using primers US18stopFOR: 5'- CGACGCCTACCTTA GACCGACAGCGGTCGTAAGCGGCAGC**TAA**GGCGACACCG CCTCCGTCTCCGAACACCATGAGTCGGC*AGGATGACGACGA TAAGTAGGG*-3' and US18stopREV: 5'-GCGGCACGATGGTGAC CGTCGGCGACTCATGGTGTTCGGAGACGGAGGCGGTGTCG CC**TTA**GCTGCCGCTTACGACCGCTG*CAACCAATTAACCAAT TCTGATTAG*-3' to replace the start codon (ATG) with a stop codon (TAA, bold) at nucleotide positions 11,150 to 11,152 (accession #EF999921). HCMV-specific sequences are underlined. For construction of the recombinant HCMV US20stop, a linear PCR product was generated using primers US20stopFOR: 5'- AGA GAAGGGTAGGTGCGCCGCAGCGGCTTTGTGCCGAGACGGT CGCCACC**TAA**CAGGCGCAGGAGGCTAACGC*AGGATGACGA CGATAAGTAGGG*-3' and US20stopREV: 5'- CCATGCGGGAGA GCAGCAGCGCGTTAGCCTCCTGCGCCTG**TTA**GGTGGCGAC CGTCTCGGCACAAAGCCGCTG*CAACCAATTAACCAATTCTG ATTAG*-3' to replace the start codon (ATG) with a stop codon (TAA, bold) at nucleotide positions 12,821 to 12,823 (accession #EF999921). HCMV-specific sequences are underlined. For construction of the HCMV US18/20stop virus, the HCMV US18stop BAC was further manipulated and the US20stop mutation was incorporated by *en passant* mutagenesis. For construction of the recombinant HCMV US18-V5, a linear PCR product was generated using primers US18-V5-FOR: 5'- CGGCGGACGCATCGACGGC GTGAGTCTCCTCAGCTTGTTG**GGTGGCGGAGGTTCGGGCA AACCGATTCCGAACCCGCTGCTGGGCCTGGATAGCACC**T AAAAAGATGGGGGAGATGA*AGGATGACGACGATAAGTAGG G*-3' and US18-V5-REV: 5'- TCTAAGCGAGCGGGAGCGTGTCA TCTCCCCCATCTTTTTA**GGTGCTATCCAGGCCCAGCAGCG GGTTCGGAATCGGTTTGCCCGAACC**TCCGCCACCCAACAA GCTGAGGAGACTCA*CAACCAATTAACCAATTCTGATTAG*-3' to introduce a C-terminal V5-tag with a GGGS linker (bold) in between nucleotide position 10,330 and 10,331 (accession #EF999921). HCMV-specific sequences are underlined. For the construction of the recombinant HCMV US20-HA, a linear PCR

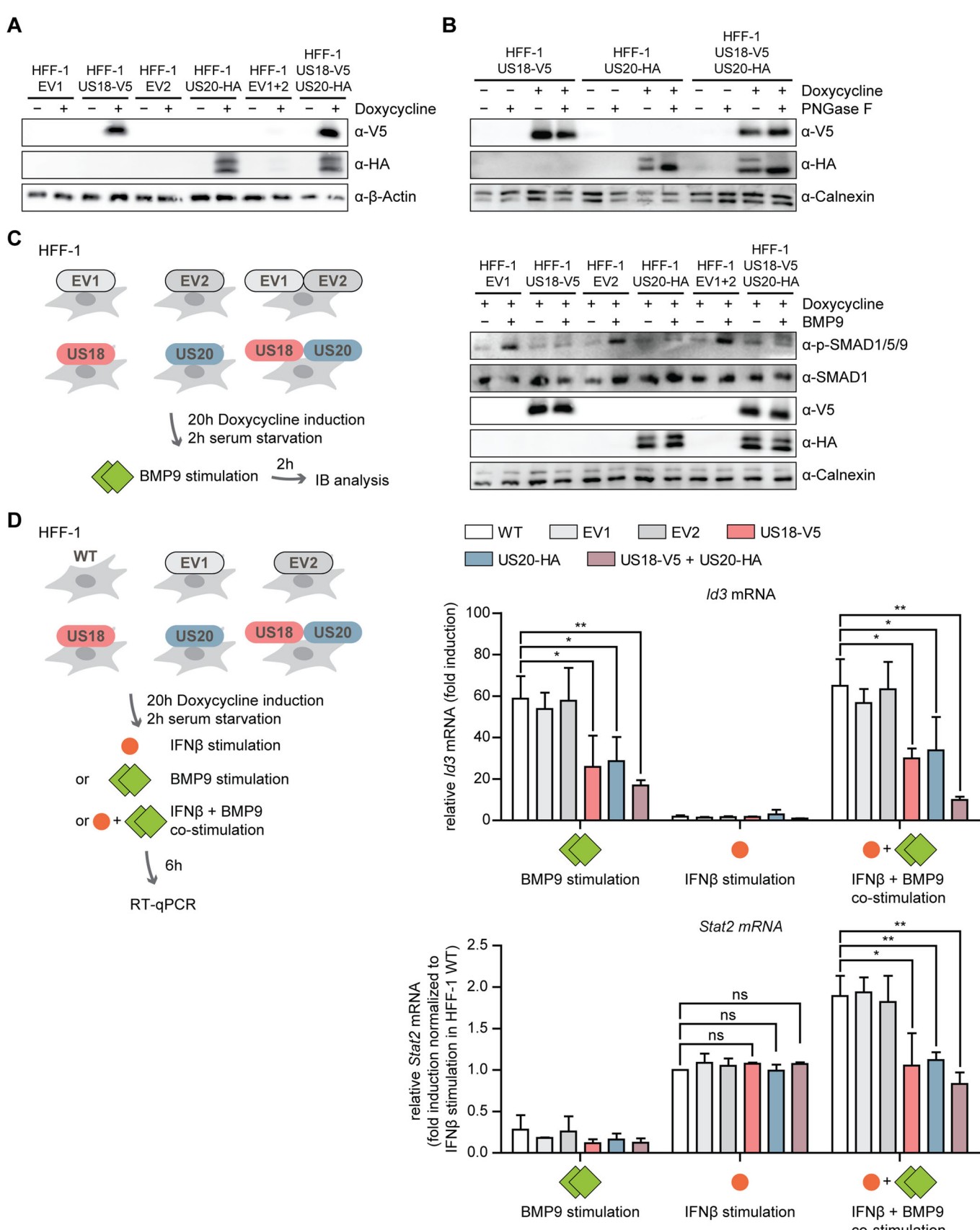

**Figure 7. Ectopically expressed HCMV US18 and US20 dampen BMP9-mediated enhancement of IFNAR signaling.**

(A) HFF-1 with doxycycline-inducible expression of US18-V5, US20-HA, or US18-V5 and US20-HA with corresponding control cell lines (EV1, EV2, EV1 + 2) were generated. Protein expression was induced with 1 µg/ml doxycycline for 20 h prior stimulation and verified by immunoblot with V5- and HA-specific antibodies, and β-Actin as a loading control. (B) HFF-1 US18-V5, HFF-1 US20-HA, or HFF-1 US18-V5 US20-HA were left untreated or protein expression was induced with 1 µg/ml doxycycline to the medium for 20 h. Cell lysates were either left untreated or treated with PNGase F for 3 h at 37 °C, followed by immunoblot analysis with V5-, HA-, and Calnexin-specific antibodies. (C, D) Indicated HFF-1 lines were treated with 1 µg/ml doxycycline to induce protein expression for 18 h, followed by 2 h of serum starvation. (C) Cells were stimulated for 1 h with BMP9 (3 nM), followed by cell lysis and immunoblot analysis with phospho-SMAD1/5/9, SMAD1, V5, HA, and Calnexin antibodies. (D) Cells were stimulated for 6 h with either IFNβ (5 ng/ml) or BMP9 (3 nM) alone, or co-stimulated with IFNβ and BMP9, followed by RNA extraction from cell lysates and RT-qPCR for *Id3* and *Stat2* transcripts. Data information: (A–C) Experiment was performed three independent times, one representative is shown. (D) Data are combined from three independent experiments. Student's *t* test (unpaired, two-tailed), n.s. not significant, *$P < 0.05$, **$P < 0.01$. Data are shown as mean ± SD. Source data are available online for this figure.

product was generated using primers US20-HA-FOR: 5'- CAACGG GACTTTGACAGCCGCCAGTACGACGGGGAAGTCC**GGTGGC GGAGGTTCGTATCCGTATGATGTGCCGGATTATGCG**TAA TGCCTATAAAACCGCGC*AGGATGACGACGATAAGTAGGG*-3' and US20stopREV: 5'- CGCGGCTGCTGTGAAAACGGGCGC GGTTTTTATAGGCATTA**CGCATAATCCGGCACATCATACGG ATACGAACCTCCGCCACC**GGACTTCCCCGTCGTACTGGG*CA ACCAATTAACCAATTCTGATTAG*-3' to introduce a C-terminal HA-tag with a GGGS linker (bold) in between nucleotide position 12,061 and 12,062 (accession #EF999921). HCMV-specific sequences are underlined. Genome regions of the recombinant HCMV BAC US18stop, US20stop and US18/20stop were amplified and sequenced to verify the correct replacement of respective start codons, and no further addition of errors were introduced compared to the HCMV TB40/E WT BAC. Recombinant HCMV BACs were reconstituted and purified as described above.

## Biosafety

Biosafety-relevant experiments with viruses were approved by the labor inspectorate (Staatliches Gewerbeaufsichtsamt) Braunschweig, Lower Saxony, Germany, and controlled by the Occupational Safety Department of TU Braunschweig in accordance with the Occupational Health and Safety Act.

## Plasmids

HCMV US18 and US20 were amplified from the HCMV TB40/E BAC (accession #EF999921) and subcloned into pcDNA3.1-V5/His (Invitrogen) via the *KpnI/NotI* sites to generate pcDNA3.1 US18-V5/His and pcDNA3.1 US20-V5/His, respectively. psPAX2, pCMV-VSV-G, and pW-TH3 (clone 10), which contain a doxycycline-inducible promoter for expression of the respective insert and a puromycin selection cassette, were kindly provided by Thomas Hennig (University of Wuerzburg, Germany). To generate pW-YC1, the puromycin selection cassette in pW-TH3 was replaced by a hygromycin selection cassette previously amplified from pMSCVhygro (Clontech). To generate pW-TH3 US18-V5, US18 was amplified from the HCMV TB40/E BAC (accession #EF999921) with oligos introducing a C-terminal V5-tag with a GGGS linker, and subcloned into the *NheI/EcoRI* sites of pW-TH3. pW-YC1 US20-HA was generated by amplification of US20 from the HCMV TB40/E BAC (accession #EF999921) with oligos introducing a C-terminal HA-tag with a GGGS linker and subcloning into *NheI/EcoRI* sites of pW-YC1. Expression constructs for M35-V5/His and M27-V5/His (both in pcDNA3.1-V5/

His, Invitrogen) have been described previously (Munks et al, 2006). pEF1α-Renilla, which expresses renilla luciferase under control of the EF1α promotor, pGL3basic MX1-Luciferase, which expresses firefly luciferase under control of the murine MX1 promoter, and pCAGGS Flag-RIG-I N, expressing a constitutively active truncation mutant of RIG-I, were kindly provided by Andreas Pichlmair (Technical University Munich, Germany). The reporter plasmid pGL3basic IFNβ-Luc (IFNβ-Luc), expressing the firefly coding sequence under the control of the murine IFNbeta enhancer, was described previously (Chan et al, 2017). pGL3 BRE Luciferase (BRE-Luc), expressing the firefly coding sequence under control of a synthetic promoter containing SMAD-binding elements (SBEs) was obtained from AddGene (#45126). pEFBOS mCherry-mSTING expressing monomeric Cherry fused to the N-terminus of murine STING, and pIRESneo3 cGAS-GFP (GFP fused to the C-terminus of human cGAS) were kindly provided by Andrea Ablasser (Global Health Institute, Ecole Polytechnique Fédérale de Lausanne, Switzerland). pIRES2-GFP was purchased from Clontech. CMVBL IRF3-5D codes for human IRF3 containing five amino acid substitutions (S396D, S398D, S402D, S404D, S405D) which render it constitutively active and was provided by John Hiscott (Institut Pasteur Cenci Bolognetti Foundation, Rome, Italy). All constructs were verified by sequencing. Primer sequences, as well as sequences of all constructs, are available upon request.

## Antibodies and reagents

Rabbit polyclonal anti-STAT1 (#9172), rabbit monoclonal anti-phospho-STAT1 (#7649, clone D4A7), rabbit monoclonal anti-SMAD1 (#6944, clone D59D7), rabbit monoclonal anti-phospho-SMAD1/5/9 (#13820, clone D5B10), rabbit monoclonal anti-SMAD2 (#5339, clone D43B4), rabbit monoclonal anti-phospho-SMAD2 (#18338, clone E8F3R), rabbit polyclonal anti-p38 (#9212), rabbit monoclonal anti-phospho-p38 (#4631, clone 12F8), rabbit monoclonal anti-p44/42 (#4695, clone 137F5), rabbit monoclonal anti-phospho-p44/42 (#4377, clone 197G2), rabbit monoclonal anti-HA (#3724, clone C29F4), and rabbit monoclonal anti-GAPDH (#2118, clone 14C10) were obtained from Cell Signaling. Mouse monoclonal anti-β-actin (A5441, clone AC-15) and rabbit polyclonal anti-Calnexin (#C4731) were obtained from Sigma-Aldrich. Mouse monoclonal anti-V5 (BLD-680605, clone 7/4) and rabbit polyclonal anti-V5 (BLD-903801) were obtained from BioLegend. Mouse monoclonal anti-ICP36 (anti-UL44) (#MBS530793, clone M612460) was purchased from MyBioSource. Mouse monoclonal anti-UL35 was described previously (Fabits

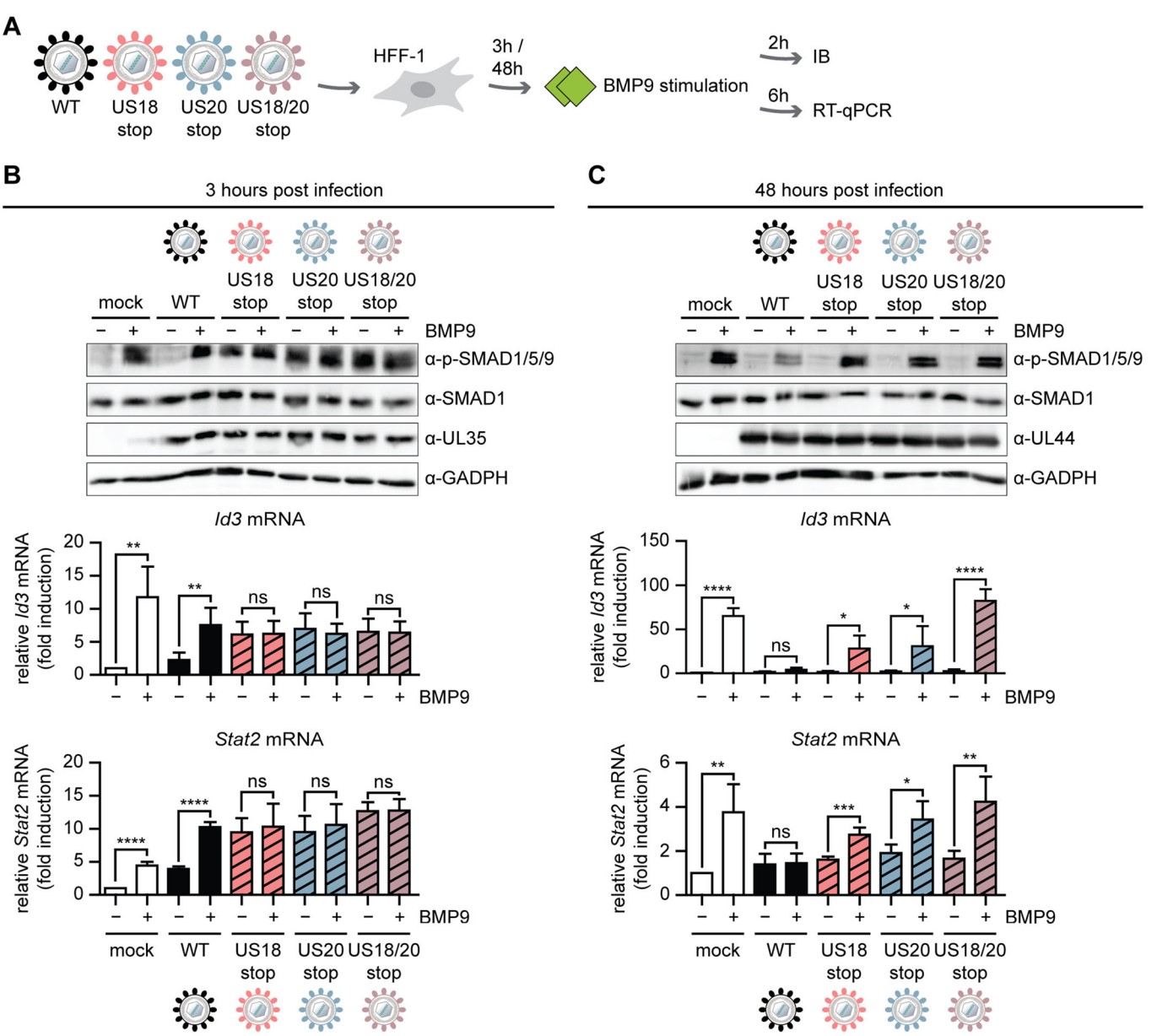

**Figure 8. HCMV US18 and US20 inhibit BMP9-mediated signaling during infection.**

(A) Schematic representation of the workflow. Recombinant HCMV US18stop, HCMV US20stop, and HCMV US18/20stop were constructed by introducing a 16 base pair (bp) stop cassette within the respective coding region. HFF-1 were infected by centrifugal enhancement with HCMV WT, HCMV US18stop, HCMV US20stop, or HCMV US18/20stop (MOI 4 for the 3 h time point, MOI 0.5 for the 48 h time point). Three or 48 h post infection, cells were stimulated with BMP9 (3 nM) for (1) 2 h, followed by cell lysis and immunoblot analysis, or (2) 6 h, followed by RNA extraction from cell lysates and RT-qPCR. (B, C) Cell lysates from (A) were subjected to immunoblot analysis with p-SMAD1/5/9, SMAD1, HCMV UL35, and GAPDH-specific antibodies (top panel), and transcript levels of *Id3* and *Stat2* are shown below. Data information: (B, C) Data are combined from three independent experiments. Student's *t* test (unpaired, two-tailed), n.s. not significant, *P < 0.05, **P < 0.01, ***P < 0.001, ****P < 0.0001. Data are shown as mean ± SD. Source data are available online for this figure.

et al, 2020). Rabbit polyclonal anti-UL82/pp71 (Tavalai et al, 2008) was kindly provided by Thomas Stamminger (Institute of Virology, Ulm University Medical Center, Ulm, Germany). Mouse monoclonal anti-IE1 (clone 63-27, originally described in (Andreoni et al, 1989) was a kind gift from Jens von Einem (Institute of Virology, Ulm University Medical Center, Ulm, Germany). Mouse monoclonal anti-ALK1 (#MAB370, clone 117702), anti-activin RIA/ALK2 (#MAB637, clone 71412) and goat polyclonal anti-BMPR-

IA/ALK3 (#AF346) were purchased from Novus Biologicals. Mouse monoclonal anti-BMPR2 (#ab130206, clone 1F12) was purchased from Abcam. Alexa Fluor®-conjugated secondary antibodies were purchased from Invitrogen. The transfection reagents Lipofectamine 2000, FuGENE HD and jetPEI were purchased from Life Technologies, Promega and Polyplus, respectively. Polybrene was obtained from Santa Cruz Biotechnology and Doxycycline was obtained from Sigma-Aldrich. OptiMEM and SYTO™ 59 red

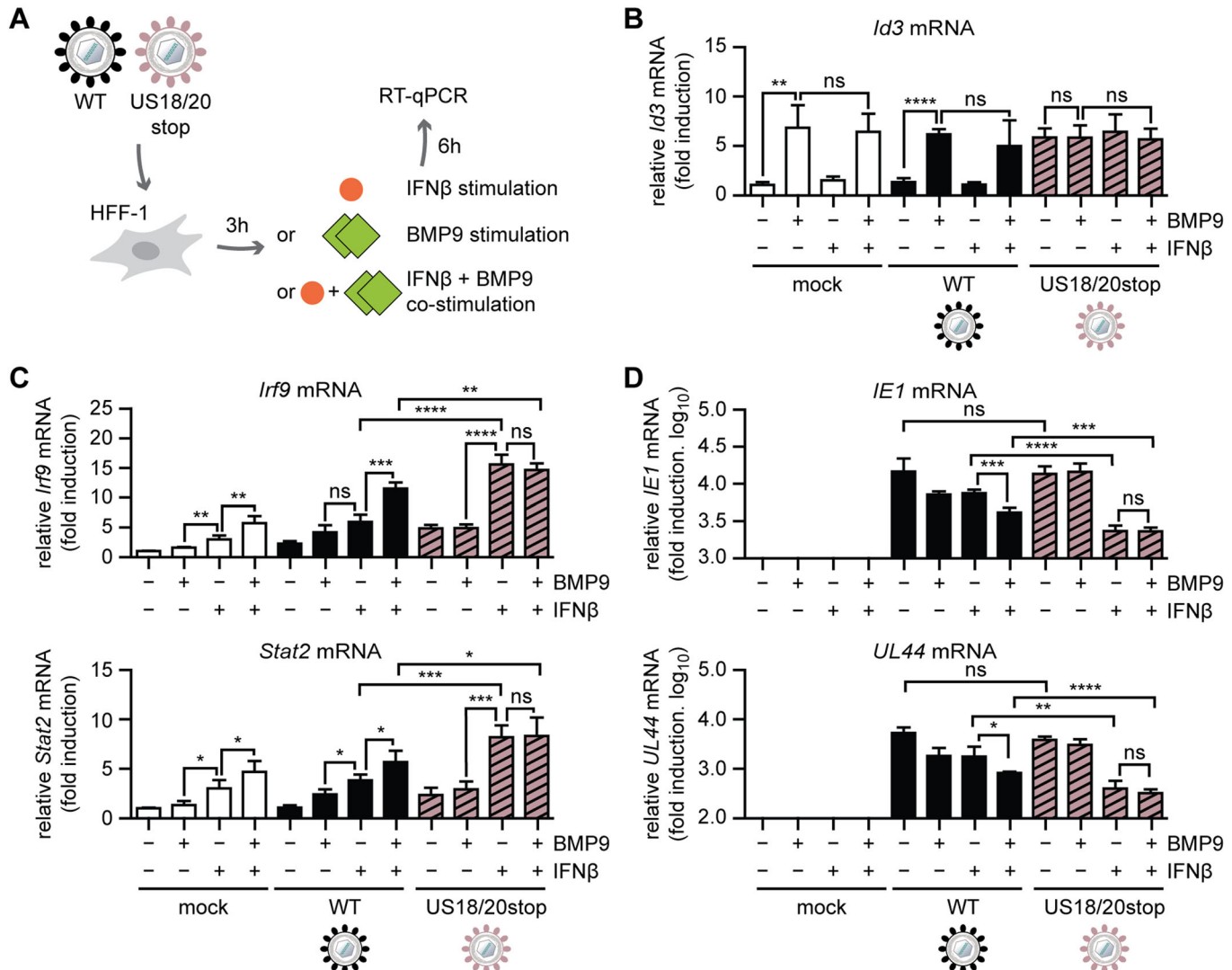

**Figure 9. HCMV lacking US18 and US20 is more sensitive to IFNβ treatment at early stages of infection.**

(A) HFF-1 were infected by centrifugal enhancement with HCMV WT or HCMV US18/20stop (MOI 4). Three hours post infection, cells were stimulated with IFNβ (1 ng/ml) or BMP9 (3 nM) alone, or co-stimulated with IFNβ and BMP9 for 6 h, followed by RNA extraction from cell lysates and RT-qPCR. (B–D) Transcript levels of *Id3* (B), the ISGs *Irf9* and *Stat2* (C), and HCMV transcripts HCMV *IE1* and HCMV *UL44* (D). Data information: Three independent experiments with similar results were performed and data shown is combined from two of the three independent experiments. Student's *t* test (unpaired, two-tailed), n.s. not significant, *$P < 0.05$, **$P < 0.01$, ***$P < 0.001$, ****$P < 0.0001$. Data are shown as mean ± SD. Source data are available online for this figure.

fluorescent nucleic acid stain were purchased from Thermo Fisher Scientific. Protease inhibitors (#4693116001) and phosphatase inhibitors (#4906837001) were purchased from Roche. Recombinant human BMP4 (#314-BP-010), human BMP6 (#507-BP-020), human BMP9 (#3209-BP-010), human BMP15 (#5096-BM-005), human Activin B (#659-AB-005), and the human/mouse anti-BMP9 antibody (#AF3209) were purchased from R&D Systems. Effective concentrations for each BMP and Activin used in this study differ and are based on the information stated on the respective datasheet. Recombinant human IFNβ was purchased from PeproTech (#300-02BC), recombinant IFNα2 (#592702) and recombinant IFNγ (#570202) were obtained from BioLegend. PNGase F (#P0704S) was obtained from New England Biolabs, and DMH1 (#S7146) and Ruxolitinib (#S1378) from Selleckchem.

## Quantitative RT-PCR

Levels of BMP type I and type II receptor transcripts were determined using 100,000 HFF-1 or 293T, respectively. Cell lysis, RNA extraction, and RT-PCR were performed as described below. For experiments with ligand stimulations, HFF-1 were seeded the day before the experiment (100,000 cells per well in a 24-well format). Two hours prior stimulation, medium was replaced by HFF-1 growth medium lacking FCS (hereinafter referred to serum starvation medium) in order to induce serum starvation of the cells, followed by the addition of the stimuli BMP4 (final concentration 18 nM), BMP6 (final concentration 18 nM), BMP9 (final concentration 3 nM), BMP15 (final concentration 18 nM), Activin B (final concentration 4 nM), IFNα2 (final concentration 5 ng/ml), IFNβ

(final concentration 5 ng/ml), or IFNγ (final concentration 5 ng/ml). After 6 h, cells were lysed in RL buffer and processed as described below. When using DMH1 or Ruxolitinib to inhibit BMP- and IFNAR-mediated signaling, respectively, the inhibitors were added to the serum starvation medium to a final concentration of 10 μg/ml after the first hour of the 2 h serum starvation step. The ligand stimulation for 6 h was performed as described above, cell lysis and processing of samples as described below. For infection experiments, HFF-1 were seeded the day before the experiment (100,000 cells per well in a 24-well format) and infected by centrifugal enhancement ($684 \times g$ at 30 °C for 45 min) the next day with HCMV WT, HCMV US18stop, HCMV US20stop, or HCMV US18/20stop (MOI 0.5 or 4). After centrifugation, cells were incubated at 37 °C for 30 min. Then, medium was exchanged to either serum starvation medium or normal growth medium for samples infected with MOI 4 or MOI 0.5, respectively. At 3 h post infection, samples infected with an MOI of 4 were stimulated with BMP9 (final concentration 3 nM) or IFNβ (final concentration 1 or 5 ng/ml) for 6 h. For samples infected with an MOI 0.5 medium was exchanged to serum starvation medium at 46 h post infection, followed by the ligand stimulation with BMP9 (final concentration 3 nM) or IFNβ (final concentration 5 ng/ml) at 48 h post infection for 6 h. Cells were lysed in RL buffer and RNA was purified using the RNA isolation kit (IST innuscreen, former AnalytikJena, #845-KS-2040250), following the manufacturer's protocol. After RNA extraction, 1500 ng of RNA per sample was used for further processing. DNase treatment and cDNA synthesis was performed with the iScript™ gDNA Clear cDNA Synthesis kit (BioRad, #1725035) following the manufacturer's protocol. Generated cDNA was diluted 1:5 before performing PCR to obtain 100 μl of cDNA. Then, RT-PCR was performed with 4 μl of cDNA per sample and the GoTaq® qPCR Master Mix (Promega, #A6001) on a LightCycler 96 (Roche). GAPDH was used for normalization. The following oligo sequences were used: Gapdh_FOR: 5'-GAAGGTGAAGGTCGGAGTC; Gapdh_REV: 5'-GAAGATGGTGATGGGATTTC; Alk1_FOR: 5'-GCGACTTCAAGAGCCGCAATGT; Alk1_REV: 5'-TAATCGCTGCCCTGTGAGTGCA; Alk2_FOR: 5'-GACGTGGAGTATGGCACTATCG; Alk2_REV: 5'-CACTCCAACAGTGTAATCTGGCG; Alk3_FOR: 5'-CTTTACCACTGAAGAAGCCAGCT; Alk3_REV: 5'-AGAGCTGAGTCCAGGAACCTGT; Alk6_FOR: 5'-CTGTGGTCACTTCTGGTTGCCT; Alk6_REV: 5'-TCAATGGAGGCAGTGTAGGGTG; Bmpr2_FOR: 5'- AGAGACCCAAGTTCCCAGAAGC; Bmpr2_REV: 5'-CCTTTCCTCAGCACACTGTGCA; Acvr2a_FOR: 5'-GCCAGCATCCATCTCTTGAAGAC; Acvr2a_REV: 5'-GATAACCTGGCTTCTGCGTCGT; Acvr2B_FOR: 5'-CGCTTTGGCTGTGTCTGGAAG; Acvr2B_REV: 5'-CAGGTTCTCGTGCTTCATGCCA; Id1_FOR: 5'-GTTGGAGCTGAACTCGGAATCC; Id1_REV: 5'-ACACAAGATGCGATCGTCCGCA; Id3_FOR: 5'-CAGCTTAGCCAGGTGGAAATCC; Id3_REV: 5'-GTCGTTGGAGATGACAAGTTCCG; Isg15_FOR: 5'-CATGGGCTGGGACCTGA; Isg15_REV: 5'-GCCGATCTTCTGGGTGATCT; Irf1_FOR: 5'-GAGGAGGTGAAAGACCAGAGCA; Irf1_REV: 5'-TAGCATCTCGGCTGGACTTCGA; Irf7_FOR: 5'-CCACGCTATACCATCTACCTGG; Irf7_REV: 5'-GCTGCTATCCAGGGAAGACACA; Irf9_FOR: 5'-CCACCGAAGTTCCAGGTAACAC; Irf9_REV: 5'-AGTCTGCTCCAGCAAGTATCGG; Ifi6_FOR: 5'-TGATGAGCTGGTCTGCGATCCT; Ifi6_REV: 5'- GTAGCCCATCAGGGCACCAATA; Stat2_FOR: 5'-CAGGTCACAGAGTTGCTACAGC; Stat2_REV: 5'-CGGTGAACTTGCTGCCAGTCTT;

Usp18_FOR: 5'-TGGACAGACCTGCTGCCTTAAC; Usp18_REV: 5'-CTGTCCTGCATCTTCTCCAGCA; Smurf1_FOR: 5'-AGTCCTCAGACACGAACTGCG; Smurf1_REV: 5'-GTCGCATCTTCATTATCTGGCGG.

## Immunoblotting

HFF-1 were seeded one day prior to the experiments (100,000 cells per well in a 24-well format). HFF-1 were washed once with serum-free medium and cultured in serum starvation medium 2 h before stimulation. Then, cells were stimulated with BMP4 (18 nM final), BMP6 (18 nM final), BMP9 (3 nM final), BMP15 (18 nM final), Activin B (4 nM final), IFNα2 (0.2, 1, or 5 ng/ml final), IFNβ (0.2, 1, or 5 ng/ml final) or IFNγ (0.2, 1, or 5 ng/ml final). Immunoblot analysis of the BMP receptors (ALK1, ALK2, ALK3, and BMPR2) was conducted in unstimulated HFF-1 and 293 T. Cells were lysed at indicated time points with radioimmunoprecipitation (RIPA) buffer (20 mM Tris–HCl pH 7.5, 1 mM EDTA, 100 mM NaCl, 1% Triton X-100, 0.5% sodium deoxycholate, 0.1% SDS). Protease and phosphatase inhibitors were added freshly to lysis buffers. Cell lysates and samples were separated by standard SDS–PAGE or Tricine-SDS-PAGE (for HCMV US18/US20 blots) and transferred to PVDF membrane (GE Healthcare) using wet transfer in Towbin blotting buffer (25 mM Tris, 192 mM glycine, 20% (v/v) methanol). Membranes were probed with the indicated primary antibodies and respective secondary HRP-coupled antibodies diluted in 5% BSA in TBS-T. Immunoblots were developed using SuperSignal West Pico or SuperSignal West Femto (Thermo Fisher Scientific) chemiluminescence substrates. Membranes were imaged with a ChemoStar ECL Imager (INTAS). Quantifications of immunoblot band intensities were performed with ImageJ. For HFF-1 lines with doxycycline-inducible expression of US18-V5 and/or US20-HA, doxycycline was added to a final concentration of 1 μg/ml 2 h post cell seeding. Eighteen hours post doxycycline treatment, medium was exchanged to serum starvation medium containing 1 μg/ml doxycycline for 2 h, followed by ligand stimulation with BMP9 (3 nM final) or IFNβ (5 ng/ml final). Effects of the inhibitors DMH1 and Ruxolitinib were tested by seeding HFF-1 the day before the experiment (100,000 cells per well in a 24-well format) and exchanging the medium to serum starvation medium containing either DMSO, Ruxolitinib (10 μM final) or DMH1 (10 μM final) for 2 h, followed by ligand stimulation with BMP4 (concentration 18 nM final) or IFNβ (5 ng/ml final) for 2 h. Cells were lysed and processed as described above. To determine the glycosylation status of US18-V5 and US20-HA, cell lysates were treated with PNGase F at 37 °C for 3 h according to the manufacturer's protocol prior to Tricine-SDS-PAGE. Protein expression in luciferase-based assays was verified by analyzing the corresponding lysates stored in 1× passive lysis buffer (Promega) by Tricine-SDS-PAGE. For immunoblotting of viral particles, 50,000 PFU per virus were diluted from the purified virus stock in RNase-free H$_2$O, SDS loading buffer was added and incubated at room temperature for 1 h, followed by separation of the proteins by Tricine-SDS-PAGE. For infection experiments, HFF-1 were seeded the day before the experiment (100,000 cells per well in a 24-well format) and infected by centrifugal enhancement ($684 \times g$ at 30 °C for 45 min) with HCMV WT, HCMV US18stop, HCMV US20stop or HCMV US18/20stop, HCMV US18-V5 or HCMV US20-HA (MOI 0.5 or 4) the next day. After centrifugation, cells were incubated at 37 °C for

30 min. Then, medium was exchanged to either serum starvation medium or normal growth medium for samples infected with MOI 4 or MOI 0.5, respectively. At 3 h post infection, samples infected with an MOI of 4 were stimulated with BMP9 (3 nM final) or IFNβ (5 ng/ml final) for 2 h. For samples infected with an MOI 0.5, medium was exchanged to serum starvation medium at 46 h post infection, followed by stimulation with BMP9 (3 nM final) or IFNβ (5 ng/ml final) at 48 h post infection for 2 h. Cells were lysed at indicated time points and processed as described above. For analysis of UL82/pp71 expression, HFF-1 were infected by centrifugal enhancement with MOI 4 of HCMV WT or HCMV US18/20stop as described above, incubated for 3 h at 37 °C, and lysed for immunoblotting.

### Immunolabeling of the HCMV IE1 antigen

HFF-1 (5000 cells per well of a 96-well plate) were seeded one day prior the experiment. The next day, medium was exchanged to serum starvation medium for 2 h, followed by stimulation with BMP4 (18 nM final), BMP6 (18 nM final), BMP9 (0.25 or 3 nM final), BMP15 (18 nM final), Activin B (4 nM final) or IFNβ (5 ng/ml final) for 6 h. Cells were then infected with HCMV at an MOI of 0.5 in normal growth medium for 16 h. Cells were washed with TBS and fixed with 4% paraformaldehyde (PFA) for 10 min. Fixed cells were permeabilized with 0.1% Triton X-100 in TBS for 5 min and blocked with TBS containing 5% FCS and 1% BSA for 30 min. Cells were immunolabeled with mouse anti-IE1 antibody, washed with TBS and labeled with secondary anti-mouse AF488 conjugated antibody. Nuclei were stained with the SYTO™ 59 red fluorescent nucleic acid stain. Red nuclei as a measurement for the total cell number and IE1 positive nuclei were counted with the IncuCyte S3 (Essen BioSciences, Sartorius, Göttingen, Germany) and the ratio of infected cells was calculated.

### Luciferase-based reporter assays

BRE- and MX1-luciferase assays: 293T (15,000 cells per well, 96-well format) were transiently transfected with 100 ng pGL3 BRE Luciferase or pGL3basic MX1-Luciferase, 1 ng pEF1α-Renilla and 120 ng plasmid of interest complexed with 0.8 µl FuGENE HD (Promega) diluted to 10 µl total volume in OptiMEM. Sixteen hours post transfection, medium was changed to serum-free DMEM to start the serum starvation process for 8 h, followed by the stimulation with BMP4 (18 nM final), BMP6 (18 nM final), BMP9 (3 nM), BMP15 (18 nM final), Activin B (4 nM final), IFNβ (5 ng/ml final) or supernatants from HCMV-infected cells (50 µl per well). To generate supernatants from HCMV-infected cells, HFF-1 were infected by centrifugal enhancement (MOI 0.5, 684 x g at 30 °C for 45 min, followed by 30 min incubation at 37 °C and replacement of virus-containing medium with fresh growth medium) with HCMV WT. For each 6 h increment, medium was exchanged to serum starvation medium for 6 h prior to harvest. Supernatants treated with the α-BMP9 antibody were first incubated with the antibody (final concentration 1 or 5 µg/ml) for 15 min prior to the 293 T stimulation. 16 h post stimulation cells were lysed in 1× passive lysis buffer (Promega).

cGAS-STING luciferase assay: 293T (25,000 cells per well, 96-well format) were transiently transfected with 60 ng pEFBOS

mCherry-mSTING, 60 ng pIRESneo3 human cGAS-GFP (for stimulated conditions), 60 ng pIRES2-GFP (for unstimulated conditions), 100 ng pGL3basic IFNβ-Luc, 1 ng pEF1α-Renilla, 120 ng plasmid of interest and 1.2 µl FuGENE HD (Promega) diluted to 10 µl total volume with OptiMEM.

RIG-I N luciferase assay: 293 T (25,000 cells per well, 96-well format) were transiently transfected with 13 ng pCAGGS Flag-RIG-I N (stimulated) or pcDNA3.1 (unstimulated) together with 50 ng pGL3basic IFNβ-Luc, 1 ng pEF1α-Renilla and 130 ng plasmid of interest and 0.66 µl FuGENE HD (Promega) diluted to 10 µl total volume with OptiMEM.

IRF3-5D luciferase assay: 293T (25,000 cells per well, 96-well format) were transiently transfected with 10 µl of FuGENE HD/DNA complexes composed of 100 ng CMVBL IRF3-5D (stimulated) or 100 ng pIRES2-GFP (unstimulated), 100 ng pGL3basic IFNβ-Luc, 1 ng pEF1α-Renilla, 120 ng plasmid of interest and 1 µl FuGENE HD (Promega) diluted to 10 µl total volume with OptiMEM.

For the cGAS-STING, RIG-I N, and IRF3-5D luciferase assays cells were lysed 20 h post transfection in 1× passive lysis buffer (Promega). Luciferase activity was measured with the dual-luciferase assay system (Promega) and an Infinite M Plex plate reader (Tecan). Luciferase fold induction was calculated by dividing Renilla-normalized values from stimulated samples by the corresponding values from unstimulated samples.

### Topology predictions and statistical analyses

Topology predictions for HCMV US18 and US20 were carried out using the online tools PredictProtein (https://predictprotein.org/), CCTOP (http://cctop.ttk.hu/) and DeepTMHMM (https://dtu.biolib.com/DeepTMHMM). Differences between the two datasets were evaluated by Student's $t$ test (unpaired, two-tailed), in the case of viral transcript levels after log transformation of the datasets, using GraphPad Prism version 5.0 (GraphPad Software, San Diego, CA). $P$ values $< 0.05$ were considered statistically significant.

## Data availability

Large primary datasets have not been generated and deposited for this study.

## Peer review information

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

## Acknowledgements

We thank Christine Standfuß-Gabisch and Maria Ebel for excellent technical assistance. This work was funded by the SMART BIOTECS alliance between the Technische Universität Braunschweig and the Leibniz Universität Hannover, an initiative supported by the Ministry of Science and Culture (MWK) of Lower Saxony, Germany (MMB), the Helmholtz Association (W2/W3-090) (MMB), and the UK Medical Research Council (MRC Human Immunology Unit core funding awarded to HD, grant no. MC_UU_12010/10).

## Author contributions

**Markus Stempel**: Conceptualization; Data curation; Formal analysis; Validation; Investigation; Visualization; Methodology; Writing—original draft; Writing—review and editing. **Oliver Maier**: Data curation; Writing—review and editing. **Baxolele Mhlekude**: Data curation; Formal analysis; Writing—review and editing. **Hal Drakesmith**: Conceptualization; Supervision; Investigation; Methodology; Writing—original draft; Writing—review and editing. **Melanie M Brinkmann**: Conceptualization; Resources; Supervision; Funding acquisition; Investigation; Methodology; Writing—original draft; Project administration; Writing—review and editing.

## Funding

## Disclosure and competing interests statement

The authors declare no competing interests. Melanie Brinkmann is an editorial advisory board member of *EMBO Reports*.

# Expanded View Figures

**Figure EV1.  Verification of BMP receptor expression and functionality of inhibitors.**

(**A**) Presence of type I (ALK1, ALK2, ALK3) and type II (BMPR2) receptors in HFF-1 and 293T was verified by immunoblotting with the respective antibodies. Detection of GAPDH protein served as loading control. (**B**) Relative transcript levels of type I (ALK1, ALK2, ALK3, ALK6) and type II (BMPR2, ACVR2A, ACVR2B) BMP receptors in 293T were determined by RT-qPCR. (**C**) 293T were co-transfected with expression plasmids for the BRE-Luciferase reporter and a Renilla luciferase normalization control (EF1α-Renilla). 24 h post transfection, 293T were either stimulated with BMP9 (3 nM), or BMP9 (3 nM) incubated for 15 min at RT with an α-BMP9 antibody (1 μg/ml or 5 μg/ml) for 16 h, followed by a dual-luciferase assay readout. (**D**) 293T were co-transfected as in (**B**). 24 h post transfection, 293 T were either stimulated with BMP4 (18 nM), BMP6 (18 nM), BMP9 (3 nM), BMP15 (18 nM), or Activin B (4 nM), or with the ligands incubated for 15 min at RT with an α-BMP9 antibody (1 μg/ml) for 16 h, followed by a dual-luciferase assay readout. (**E**) HFF-1 were either mock, DMSO, Ruxolitinib (10 μM) or DMH1 (10 μM) treated for 2 h, followed by stimulation with IFNβ (5 ng/ml) or BMP4 (18 nM) for 2 h. Cells were lysed and lysates were subjected to immunoblot analysis with p-STAT1, STAT1, p-SMAD1/5/9, SMAD1, p-p38, p38, p-p44/42, p44/42, and Calnexin-specific antibodies. Data information: (**A–E**) Experiments were performed two independent times, one representative is shown. Luciferase fold induction was calculated by dividing Renilla-normalized values from stimulated samples by the corresponding values from unstimulated samples.

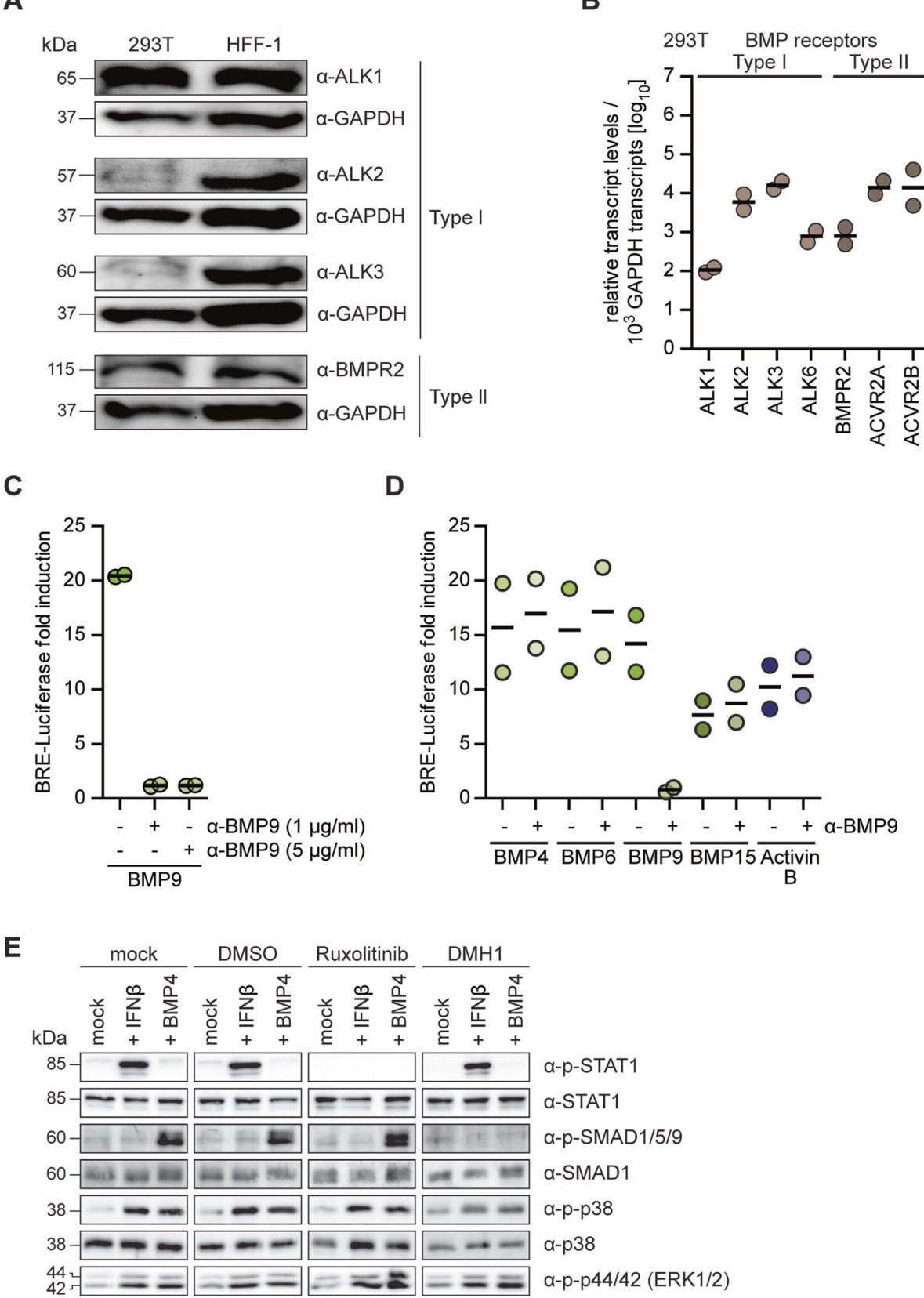

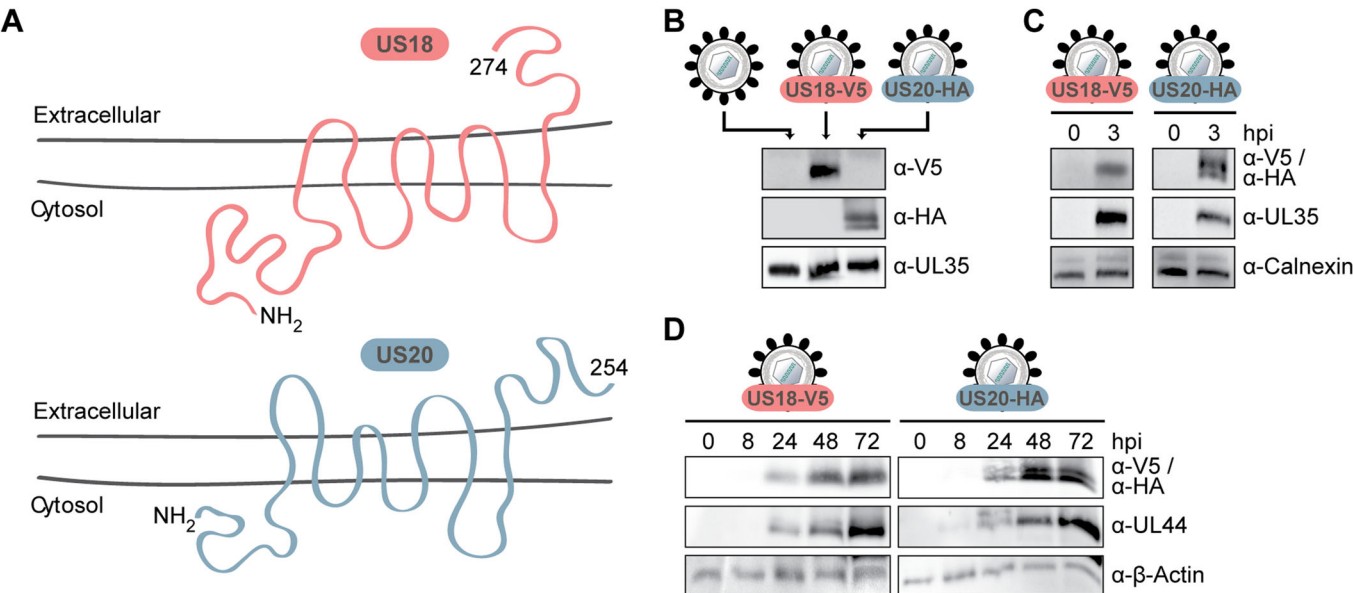

**Figure EV2.   HCMV US18 and US20 are associated with HCMV particles and are de novo expressed with early kinetics.**

(A) Topology prediction of HCMV US18 and HCMV US20. NH₂ indicates the N-terminus, and the number indicates protein length in amino acids. Topology predictions for HCMV US18 and US20 were carried out using the online tools PredictProtein (https://predictprotein.org/), CCTOP (http://cctop.ttk.hu/) and DeepTMHMM (https://dtu.biolib.com/DeepTMHMM). (B) Recombinant HCMV expressing V5-tagged US18 or HA-tagged US20 were generated. 50.000 PFU of HCMV WT, HCMV US18-V5 and HCMV US20-HA were analyzed by immunoblot with antibodies for V5, HA and the HCMV tegument protein UL35. (C) HFF-1 were infected by centrifugal enhancement with HCMV US18-V5 or HCMV US20-HA (MOI 4). 3 h later, cells were lysed and cell lysates analyzed by immunoblotting with V5-, HA-, UL35- and Calnexin-specific antibodies. (D) HFF-1 were infected by centrifugal enhancement with HCMV US18-V5 or HCMV US20-HA (MOI 0.5) for the indicated time points. Cells were lysed and lysates analyzed by immunoblot with V5-, HA-, UL44- and β-Actin-specific antibodies. Data information: (B–D) Experiment was performed three independent times, one representative is shown.

                    

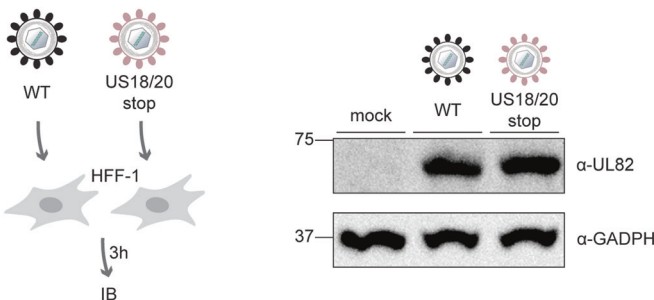

**Figure EV3. The HCMV US18/20stop mutant infects HFF-1 with similar efficiency as HCMV WT.**

HFF-1 were infected by centrifugal enhancement at MOI 4 with HCMV WT or HCMV US18/20stop and lysed 3 h later. Expression of the HCMV tegument protein UL82/pp71 was analyzed by immunoblotting with a UL82/pp71-specific antibody. GAPDH served as loading control. Data information: One representative of two independent experiments is shown.

