## [Peer Review File · EMBO Reports]

Novel role of bone morphogenetic protein 9 in innate host responses to HCMV infection

Melanie Brinkmann, Markus Stempel, Hal Drakesmith, Baxolele Mhlekude, and Oliver Maier

Corresponding author(s): Melanie Brinkmann (m.brinkmann@tu-bs.de)

Review Timeline:

Submission Date:	24th Mar 23
Editorial Decision:	14th Apr 23
Revision Received:	3rd Sep 23
Editorial Decision:	3rd Oct 23
Revision Received:	3rd Jan 24
Accepted:	17th Jan 24

Editor: Achim Breiling

Transaction Report:

Dear Prof. Brinkmann,

Thank you for the submission of your manuscript to EMBO reports. I have now received the reports from the three referees that were asked to evaluate your study, which can be found at the end of this message.

As you will see, the referees state that these findings are of high interest. However, they have several comments, concerns, and suggestions, indicating that a major revision of the manuscript is necessary to allow publication of the study in EMBO reports. As the reports are below, and all the referee concerns need to be addressed as indicated in the reports, I will not detail them here.

Given the constructive referee comments, I would like to invite you to revise your manuscript with the understanding that all referee concerns must be addressed in the revised manuscript and in a detailed point-by-point response. Acceptance of your manuscript will depend on a positive outcome of a second round of review. It is EMBO reports policy to allow a single round of revision only and acceptance of the manuscript will therefore depend on the completeness of your responses included in the next, final version of the manuscript.

- 1) a .docx formatted version of the final manuscript text (including legends for main figures, EV figures and tables), but without the figures included. Figure legends should be compiled at the end of the manuscript text.
- 2) individual production quality figure files as .eps, .tif, .jpg (one file per figure), of main figures (up to 8) and EV figures. Please upload these as separate, individual files upon re-submission.

- 3) a complete author checklist, which you can download from our author guidelines (<https://www.embopress.org/page/journal/14693178/authorguide>). Please insert page numbers in the checklist to indicate where the requested information can be found in the manuscript. The completed author checklist will also be part of the RPF.

- 4) that primary datasets produced in this study (e.g. RNA-seq, ChIP-seq, structural and array data) are deposited in an appropriate public database. If no primary datasets have been deposited, please also state this in a dedicated section (e.g. 'No primary datasets have been generated and deposited'), see below.

The accession numbers and database should be listed in a formal "Data Availability" section (placed after Materials & Methods) that follows the model below. This is now mandatory (like the COI statement). Please note that the Data Availability Section is restricted to new primary data that are part of this study. This section is mandatory. As indicated above, if no primary datasets have been deposited, please state this in this section

Data availability

5) We now request the publication of original source data with the aim of making primary data more accessible and transparent to the reader. Our source data coordinator will contact you to discuss which figure panels we would need source data for and will also provide you with helpful tips on how to upload and organize the files.

6) Our journal encourages inclusion of *data citations in the reference list* to directly cite datasets that were re-used and obtained from public databases. Data citations in the article text are distinct from normal bibliographical citations and should directly link to the database records from which the data can be accessed. In the main text, data citations are formatted as follows: "Data ref: Smith et al, 2001" or "Data ref: NCBI Sequence Read Archive PRJNA342805, 2017". In the Reference list, data citations must be labeled with "[DATASET]". A data reference must provide the database name, accession number/identifiers and a resolvable link to the landing page from which the data can be accessed at the end of the reference. Further instructions are available at: <http://www.embopress.org/page/journal/14693178/authorguide#referencesformat>

7) Regarding data quantification and statistics, please make sure that the number "n" for how many independent experiments were performed, their nature (biological versus technical replicates), the bars and error bars (e.g. SEM, SD) and the test used to calculate p-values is indicated in the respective figure legends (also for potential EV figures and all those in the final Appendix). Please also check that all the p-values are explained in the legend, and that these fit to those shown in the figure. Please provide statistical testing where applicable. Please avoid the phrase 'independent experiment', but clearly state if these were biological or technical replicates. Please also indicate (e.g. with n.s.) if testing was performed, but the differences are not significant. In case n=2, please show the data as separate datapoints without error bars and statistics. See also: <http://www.embopress.org/page/journal/14693178/authorguide#statisticalanalysis>

8) Please also note our reference format:

9) We updated our journal's competing interests policy in January 2022 and request authors to consider both actual and perceived competing interests. Please review the policy <https://www.embopress.org/competing-interests> and update your competing interests if necessary. Please name this section 'Disclosure and Competing Interests Statement' and put it after the Acknowledgements section.

10) We now use CRediT to specify the contributions of each author in the journal submission system. CRediT replaces the author contribution section. Please use the free text box to provide more detailed descriptions. See also guide to authors: <https://www.embopress.org/page/journal/14693178/authorguide#authorshippinguidelines>

11) Please add up to 5 keywords to the manuscript text below the abstract and order the manuscript sections like this, using these names:

Title page - Abstract - Keywords - Introduction - Results - Discussion - Materials and Methods - Data availability section - Acknowledgements - Disclosure and Competing Interests Statement - References - Figure legends - Expanded View Figure legends

I look forward to seeing a revised version of your manuscript when it is ready. Please let me know if you have questions or comments regarding the revision.

Yours sincerely,

Referee #1:

Stempel M. et al demonstrate a new role of BMP9 in the context of HCMV infection in particular and BMP9 in modulating the innate antiviral immune response in general. They show that BMP9 blocks HCMV in an IFNAR signaling-enhancement way, mainly by co-operation with type I interferons. In addition, they identify the viral proteins US18, US20 as inhibitors of the BMP9 antiviral activity and they nicely verify the co-stimulation activity of BMP9 and IFN I for a BMP-9 dependent HCMV suppression.

Although the authors did a really good job in the logical explanation of their study and experiments, there were some points they need to further elaborate on.

First, it was not justified why BMP15 was not a promising hit and was excluded from further investigation. The authors claim that BMP15 and activin A are negative regulators of cell proliferation. On contrary, Qin et al. show that Knock Down of BMP15 had a negative effect on cell proliferation and growth (fig 7D) suggesting that BMP15 has rather a positive impact. Respectively, Zaragosi et al. show that "activin A supplementation enhanced cell number in proliferating hMADS cell cultures" indicating the opposite of authors claim.

Second, in a lot of graphs, the y axis labeling gives "relative" values, but it is not clear from the legend, nor the text to what is normalized or referenced to. Therefore, since absolute numbers are missing, it is difficult to assess the data.

Third, in some graphs data from two independent experiments are combined, yet the authors do statistical analysis. This is not a sound statistical approach. I would suggest either replicating those experiments once more or remove the statistical analysis for those experiments since at least 3 replicates are needed to perform a proper statistical analysis.

Figure 2: A) Since it is not clear to what those levels are relative, we can't conclude if the expression of the receptors is high or low in HFFs. B) Its not completely clear, why different concentrations of the BMPs were used.

Figure 3: B) The concentrations of the stimuli used are not mentioned. Perhaps it should be explained, that the concentrations are the same as Figure 2, if this was the case.

Figure 4: C) Could be explained why the fold induction of ISGs is different to the one in figure 2? In some Cases (Irf9) it is 8 times lower compared to the levels in fig2.

I believe that the addition of a negative control would have been helpful in this experiment (for example a different BMP) to show that this co-activity is specific for BMP9.

The last two graphs are inconsistently y-labeled compared (different normalization) to the previous 6 graphs in figure 4C. Is there a reason why?

E) Could you please explain why the combination of IFN β and BMP9 here slightly increases Stat2 mRNA compared to IFN β alone, while in 4C this combination resulted in more prominent induction?

Figure 5: The authors did not check the phosphorylation of STAT2 as well. Is there a reason why they didn't investigate this along with STAT1?

Figure 6: B) Is there an explanation why there is a significant increase in MX1 luciferase activity when cotransfecting with US18 or US20 and stimulate with IFN β only? We also see an increased activity with US20 upon cotreatment with IFN β and BMP9, compared to EV, although one would expect decreased activity.

E) The expression of M35 and the other 2 proteins is not comparable. Could that mask some effects of the proteins?

Figure 8) B) Since BMP9 stimulates the IFNAR signaling mainly in combination with IFN type I, it would make sense to set up this experiment with co-incubation similar to figure 7. In this way, it could be assessed if KO of US18 and US20 completely rescues the response that occurs from the co-stimulation or if there are other HCMV factors/transcripts that play a role in this mechanism.

C) It is evident from figure 3D that the BMP pathway was similarly activated 0-6 hpi and 42-48 hpi. Since US18 and US20 are knocked out wouldn't we expect high phosphorylation of SMAD1/5/9 in the samples where cells were infected with the mutant viruses, even without the addition of BMP9? The infection itself should be enough to increase BMP9, comparably to 3 hours post infection.

Figure 9) D) The authors nicely demonstrate the co-operation of BMP-9 and IFN β against HCMV: However, treatment with BMP9 only resulted in lower expression of IE1, similar to treatment with only IFN β , indicating that BMP9 alone harbors an antiviral activity, something that is not consistent with figure 3b.

Abstract:

Line 9: I wouldn't use the word 'replication' since the authors didn't specifically investigate different parts of the viral cycle (ie, entry, egress, etc)

Discussion:

Line 304-305: 'which relatively specifically activate', could you rephrase this please?

Methods:

Line 357: HFF-1 is rather a "primary cell line", generated from a pool of 2 donors.

Line 375: For how long did the selection procedure last?

Referee #2:

In this manuscript, Stempel et al report a link between bone morphogenetic protein 9 (BMP9) and HCMV infection with respect to BMP9-induced type I (IFN) signaling and subsequent avoidance of this anti-viral response by the virus. In detail, they show that HCMV infection of fibroblasts results in increased levels of BMP9 which then increases the antiviral activity of IFN β to help restrict HCMV replication. However, they also show that viral US18 and US20 counteract this, likely by these viral genes downregulating type I BMP receptors.

As would be expected from these authors, the data presented are of high quality and result from a comprehensive set of experiments which appear to have been carefully carried out and sensibly interpreted.

That said, the authors should address the following:

1) Figure 2A. Whilst I accept that levels of RNAs of these receptors will likely reflect levels of protein, the authors should consider confirming protein levels for key receptors in their study (antibodies are available for many of them).

2) Figure 3A/B. It is clear that BMP9 stimulation by itself has little effect on IE1 expression, but BMP9 + IFN β has more of an effect than IFN β alone. An important question is at what level is this acting - the assay appears to be an analysis of IE levels after an overnight infection (16h post-infection). Is this at the level of virus "uptake" and is it at the level of viral IE transcription (RNA levels)?

3) In figure 3A/B, the authors also suggest that the ability of BMP15 or Activin B, alone, to limit IE expression might be due to suppression of fibroblasts proliferation - do they have any evidence that this is the case as this seems counterintuitive to this reviewer on the basis that cells in S-phase or G2/M support IE expression poorly compared to cells in e.g. G0/G1?

4) Figure S1A, again, have the authors considered confirming protein levels for these receptors?

5) Figure S1C - fold induction over what?

6) Figure 3D. I see no primary data that shows that antibodies to BMP4/6/15/Activin B block infected HFF supernatants from activating BRE-Luc - this should be shown or stated.

7) Figure 6B. Overexpression of US18 or US20 (or together) dampens the ability of BMP9 to activate BRE-Luc. Have the authors confirmed that this is as a result of down-regulation of BMP receptors in their reporter cell line?

Additionally, (figure 6B, lower panel) why does US20 (but not US18) augment IFN β activation of MX1-Luc (and what happens if both are co-expressed)? And, on this basis, is the statement that US18/US20 "...do not affect IFNAR signaling" (line 211) correct - or consistent with the view, throughout, that the absence of US20 increases sensitivity to IFN β ?

8) Figure 8. Whilst I accept the authors believe that the effects of US18 and US20 stop viruses are due to the inability of these viruses to down-regulate BMP receptors, have they ruled out additional effects such as decreased BMP9 induction - do effects on Luc expression in their BRE-Luc reporter cells differ if they use supernatants from WT/US18stop/US20stop virus?

9) Finally, Pham et al (2021) have shown that TGF β , also induced upon lytic infection, limits induction of type I interferons. This should at least be mentioned.

Referee #3:

This study demonstrates that BMP9 plays a role in the antiviral response to HCMV, partly through interfacing with the IFN response, and that HCMV antagonizes this response through the US18 and US20 proteins. This information is interesting, and it is very clear that the authors have discovered some novel biology relating to BMP9 function and its antiviral activity, as well as identifying HCMV genes that antagonize these processes. This certainly impacts our understanding of HCMV, and may be relevant to other viruses too. However the study also has several limitations. My main two criticisms are that:

1. The study is very correlative, and although many of these correlations are consistent with the hypothesis put forward, they stop short of actually proving the mechanistic links that they claim.

2. The vast majority of the work uses a huge amount of biochemical and cell biology analysis to draw conclusions about the signalling pathways involved. What really matters at the end of the day is the effect of these pathways on the virus - i.e. do these pathways result in better viral control. And here the data could be far stronger - the use of IE1 or UL44 mRNA levels as a readout for 'antiviral capacity' is unusual, and (in my opinion) not sufficient.

My specific points in more detail are:

Fig 3B. At a MOI=0.5, how come all cells are infected? Would they see a stronger antiviral impact if they were infecting cells at a level just below 100%?

Figure 5A - the effects of BMP9 on STAT1 phosphorylation are very weak, yet they claim that this is the likely mechanism by which BMP9 affects IFN signalling. Are those changes really sufficient to make that claim? Can they formally show that this conclusion is mechanistically true?

Fig 9. A key claim of the paper is that targeting of ALK1/2 by US18/US20 is responsible for the effects seen. Yet in the referenced paper, the level of ALK1/2 recovery in the delta US18/20 mutant looks very minimal, and the addition of BMP9 has clear effects against the WT virus, when (assuming this virus has targeted ALK1/2), it shouldn't be. Can the authors show by flow the levels of ALK1/2 in these mutants, at both early and late times, in order to support their conclusions? Is there a way to modulate ALK1/2 downregulation (e.g. by overexpression/knockout) to definitively show that this is how US18/20 are blocking BMP9 signalling?

Along related lines, the authors use a single stop codon in place of the start codon to knock out their genes - there are good reasons for choosing this modification, but it comes with the risk of residual translation. Can they show by western blot that this results in a complete knockout of US18/20 expression?

Fig 9. I'd expect the ability of US18/20 to target BMP9 mediated antiviral effects to be strongest at late times, following de novo expression. In that case, why is the only analysis done at early times, when US18/20 levels delivered from the virion are likely to be fairly low? Furthermore, the data here seems inconsistent with some of the core claims, with the WT showing effects that are very similar to the knockout virus in some cases, and even an enhanced antiviral effect against the WT in some cases (e.g. the effect of BMP1 on IE1 mRNA). How can the authors explain these results if their hypothesis is true? The choice to measure RNA transcripts as a readout is odd, and gives very limited information - e.g. does this reflect lower expression in the cell, or fewer infected cells? What are the knock-on consequences of this to the virus? It's also odd to choose UL44 given that I'd assume UL44 is not expressed well at 6h? I think it would really help their case if the authors could also show antiviral effects at the level of virus spread or release - i.e. adding BMP9/IFN β onto cultures of spreading virus, to formally demonstrate that these molecules and pathways have the antiviral effects they claim, and that these activities are functionally antagonized by US18/20.

Figure S2 - ideally virus needs to be purified on a glycerol/tartrate gradient before concluding that proteins are part of the virion, as opposed to cellular contamination. It would also help if a control for cellular contamination was included in the blots (e.g. calnexin?).

Referee #1:

Stempel M. et al demonstrate a new role of BMP9 in the context of HCMV infection in particular and BMP9 in modulating the innate antiviral immune response in general. They show that BMP9 blocks HCMV in an IFNAR signaling-enhancement way, mainly by co-operation with type I interferons. In addition, they identify the viral proteins US18, US20 as inhibitors of the BMP9 antiviral activity and they nicely verify the co-stimulation activity of BMP9 and IFN I for a BMP-9 dependent HCMV suppression.

Although the authors did a really good job in the logical explanation of their study and experiments, there were some points they need to further elaborate on.

We thank the reviewer for the effort and time to review our manuscript and the positive and constructive feedback.

First, it was not justified why BMP15 was not a promising hit and was excluded from further investigation. The authors claim that BMP15 and activin A are negative regulators of cell proliferation. On contrary, Qin et al. show that Knock Down of BMP15 had a negative effect on cell proliferation and growth (fig 7D) suggesting that BMP15 has rather a positive impact. Respectively, Zaragosi et al. show that "activin A supplementation enhanced cell number in proliferating hMADS cell cultures" indicating the opposite of authors claim.

Indeed, the effect of BMP15 and Activin B on HCMV is very interesting, and we are following up on this observation in a follow-up project since the aim of this study was to identify a BMP-type I IFN axis in the context of HCMV infection as already shown for other BMPs. Only BMP9 showed a co-stimulatory effect with IFN, and this is why we focused our attention on this BMP. To better explain our rationale for investigating the BMP9-IFN phenotype, we rephrased this paragraph in the results section of our manuscript. Thank you for pointing out the other two studies, we have now adjusted the corresponding paragraph and rephrased our decision to focus on the phenotype of BMP9.

Second, in a lot of graphs, the y axis labeling gives "relative" values, but it is not clear from the legend, nor the text to what is normalized or referenced to. Therefore, since absolute numbers are missing, it is difficult to assess the data.

Thank you for pointing this out. As stated in the M+M part, transcript levels were generally normalized to GAPDH transcript levels, hence relative transcript levels are presented in the graph. We have now improved our explanations in the figure legend of Figure 2B, as this is the first occurrence.

Third, in some graphs data from two independent experiments are combined, yet the authors do statistical analysis. This is not a sound statistical approach. I would suggest either replicating those experiments once more or remove the statistical analysis for those experiments since at least 3 replicates are needed to perform a proper statistical analysis.

We have removed the statistical analysis in the corresponding figures.

Figure 2: A) Since it is not clear to what those levels are relative, we can't conclude if the expression of the receptors is high or low in HFFs. B) Its not completely clear, why different concentrations of the BMPs were used.

*A) That is correct. It was not our intention to show expression levels, only to show whether the receptors are detectably expressed or not. We adjusted the regarding M+M section to clarify this.
B) The effect concentrations of the BMPs differ and we used the effect concentrations as given on the data sheets of the respective BMP and based on the literature. To clarify this, we added a sentence in the M+M section.*

Figure 3: B) The concentrations of the stimuli used are not mentioned. Perhaps it should be explained, that the concentrations are the same as Figure 2, if this was the case.

Thank you for pointing this out. We now included this information in the legend.

Figure 4: C) Could be explained why the fold induction of ISGs is different to the one in figure 2? In some Cases (Irf9) it is 8 times lower compared to the levels in fig2.

Transcript levels are normalized to a housekeeping gene, and since the method is very sensitive, even slight differences in the original concentration (e.g. dependent on the current proliferation / cell cycle state or different batches of the same cell line) are amplified and can subsequently affect fold induction levels of the gene of interest. However, because this only results in inter-assay, but not intra-assay variation, conclusions made from respective experiments are still valid and can be compared to one another.

I believe that the addition of a negative control would have been helpful in this experiment (for example a different BMP) to show that this co-activity is specific for BMP9.

As we show in Figure 3B, the co-activity is specific for BMP9 regarding HCMV. Hence, we only followed this phenotype and did not include other BMPs from Figure 3B onwards.

The last two graphs are inconsistently y-labeled compared (different normalization) to the previous 6 graphs in figure 4C. Is there a reason why?

The reason is that the fold inductions were too different and could therefore not be combined (please see our answer regarding your comment with respect to Figure 4).

E) Could you please explain why the combination of IFN β and BMP9 here slightly increases Stat2 mRNA compared to IFN β alone, while in 4C this combination resulted in more prominent induction?

Since the experiment in Figure 4E incorporates the treatment with inhibitors, which - although they have been used in various studies before and are reported to be specific - may affect other cell processes, one cannot expect the very same phenotypes as in cells which were not additionally stressed with inhibitor incubations. Thus, albeit weaker signature, the general phenotype was confirmed.

Figure 5: The authors did not check the phosphorylation of STAT2 as well. Is there a reason why they didn't investigate this along with STAT1?

In the literature, either of the two STATs have been frequently used to demonstrate activation of the IFNAR pathway and we decided for STAT1.

Figure 6: B) Is there an explanation why there is a significant increase in MX1 luciferase activity when cotransfecting with US18 or US20 and stimulate with IFN β only? We also see an increased activity with US20 upon cotreatment with IFN β and BMP9, compared to EV, although one would expect decreased activity.

Since US18 and US20 mediate downregulation of BMPR, but not of the IFNAR, they should not decrease MX1 luciferase activity (MX1 is an ISG induced by IFNAR signaling) upon IFNAR stimulation.

As the experimental setup is based on the determination of downregulation, the assay is neither designed, nor does it include any control to conclusively interpret enhancement of luciferase activity.

Thus, while we agree that this may be an interesting point, any conclusion regarding the enhanced luciferase activity in presence of US18 and/or US20 cannot be interpreted at this stage. Effectively, the aim of this experiment was to show that MX1 activity is not decreased in presence of US18/US20, showing their specificity for BMPR signaling, and this is exactly what we see.

E) The expression of M35 and the other 2 proteins is not comparable. Could that mask some effects of the proteins?

It is true that M35 expression is stronger than that for US18 and US20. We titred M35 and could show that it downmodulates PRR signaling even at very low expression levels (Schwanke et al., doi 10.1128/jvi.00400-23 Figure 3D). Nevertheless, at the given expression level of US18 and US20, they do have an effect on BMP signaling (Figure 6B upper panel), while Figure 6E clearly shows that they do not

have an effect on PRR signaling when the same amount of plasmid DNA was transfected as before and the protein levels were comparable to Figure 6B.

Figure 8) B) Since BMP9 stimulates the IFNAR signaling mainly in combination with IFN type I, it would make sense to set up this experiment with co-incubation similar to figure 7. In this way, it could be assessed if KO of US18 and US20 completely rescues the response that occurs from the co-stimulation or if there are other HCMV factors/transcripts that play a role in this mechanism.

In Figure 7 we used HFF cells expressing US18 and US20 ectopically and co-stimulated with BMP9 and IFN because the cells would otherwise not be stimulated (since we do not treat with a PRR agonist or infect with HCMV). In Figure 8, we infect with wildtype HCMV or HCMV lacking US18, US20, or both. These viruses all induce the type I IFN response – hence, it was not necessary to add exogenous IFN. In Figure 9B,C,D, we did perform such a co-stimulation with BMP9 and IFN during viral infection as suggested by the reviewer to explore whether HCMV lacking US18/US20 is more sensitive to IFN treatment.

C) It is evident from figure 3D that the BMP pathway was similarly activated 0-6 hpi and 42-48 hpi. Since US18 and US20 are knocked out wouldn't we expect high phosphorylation of SMAD1/5/9 in the samples where cells were infected with the mutant viruses, even without the addition of BMP9? The infection itself should be enough to increase BMP9, comparably to 3 hours post infection.

In Figure 8B, 3h post infection, we see exactly what the reviewer describes here: In absence of US18/US20, 3phospho-SMAD levels are higher in the absence of BMP9 stimulation, whereas at 48h post infection we only see higher 3phospho-SMAD levels in presence of exogenous BMP9. At 48 hours post infection, almost the full repertoire of the HCMV proteins is present in the infected cells, manipulating many cellular pathways to benefit replication. Since the BMP pathway is implicated in proliferation and cell migration, it is very likely that other HCMV proteins add to the phenotype. However, we do not have data currently to provide a full explanation for this observation and this interpretation.

Figure 9) D) The authors nicely demonstrate the co-operation of BMP-9 and IFN β against HCMV: However, treatment with BMP9 only resulted in lower expression of IE1, similar to treatment with only IFN β , indicating that BMP9 alone harbors an antiviral activity, something that is not consistent with figure 3b.

The two experiments listed by the reviewer have very different starting points.

In Figure 3B, we pre-stimulated the cells with BMP9 before infection in order to see if the cells can establish an antiviral state in the cell, independent of the presence of the virus. While BMP9 induces ISGs to a certain degree, this pre-stimulation with BMP9 only apparently does not provide a kickstart when cells are subsequently infected with HCMV.

In Figure 9D, the cells were first infected at a high MOI (MOI 4 vs. MOI 0.5 in Figure 3B), thus (1) presenting the virus with a different environment which is more leaned towards the endogenous secretion of BMP9 in HCMV-infected cells as shown in Figure 3D and (2) adding BMP9 to infected cells at a stage when it is naturally secreted and thus may act in a similar manner as its endogenous counterpart.

Abstract:

Line 9: I wouldn't use the word 'replication' since the authors didn't specifically investigate different parts of the viral cycle (ie, entry, egress, etc)

We agree and have omitted the word “replication”.

Discussion:

Line 304-305: 'which relatively specifically activate', could you rephrase this please?

We have changed it to “which activate”.

Methods:

Line 357: HFF-1 is rather a "primary cell line", generated from a pool of 2 donors.

Thank you – we have changed it.

Line 375: For how long did the selection procedure last?

We have added the following paragraph to the M+M section:

"Selection of cells with puromycin lasted over approx. 1.5 weeks with at least 3 passages, and selection with hygromycin over approx. 2.5 weeks with at least 5 passages. Untransduced control cells were treated with the antibiotics side-by-side to evaluate when the selection reached its endpoint. Occasionally, antibiotics were added to the medium when cells were passaged. Experiments were always performed with cells that were not treated with antibiotics for at least 1 passage prior the experiment."

Referee #2:

In this manuscript, Stempel et al report a link between bone morphogenetic protein 9 (BMP9) and HCMV infection with respect to BMP9-induced type I (IFN) signaling and subsequent avoidance of this anti-viral response by the virus. In detail, they show that HCMV infection of fibroblasts results in increased levels of BMP9 which then increases the antiviral activity of IFN β to help restrict HCMV replication. However, they also show that viral US18 and US20 counteract this, likely by these viral genes downregulating type I BMP receptors.

As would be expected from these authors, the data presented are of high quality and result from a comprehensive set of experiments which appear to have been carefully carried out and sensibly interpreted.

We thank the reviewer for these nice words and appreciate the time and effort to review our data.

That said, the authors should address the following:

1) Figure 2A. Whilst I accept that levels of RNAs of these receptors will likely reflect levels of protein, the authors should consider confirming protein levels for key receptors in their study (antibodies are available for many of them).

We agree and have now added a panel showing protein expression of the type I receptors ALK1, ALK2, and ALK3 and the type II receptor BMPR2 in HFF-1 and 293T cells (see new Supplementary Figure 1A). It took a very long time for them to arrive which caused this delay in submitting the revised version, but we completely agree that it is important to not solely rely on the expression of transcripts.

2) Figure 3A/B. It is clear that BMP9 stimulation by itself has little effect on IE1 expression, but BMPR9 + IFN β has more of an effect than IFN β alone. An important question is at what level is this acting - the assay appears to be an analysis of IE levels after an overnight infection (16h post-infection). Is this at the level of virus "uptake" and is it at the level of viral IE transcription (RNA levels)?

In Figure 3A/B we pre-stimulate HFF cells with BMP only, IFN only or BMP+IFN in combination and 6h later infect the cells with HCMV and detect HCMV IE1 protein levels by immunofluorescence 16h later. In Figure 9, the setup is different: we first infect HFF cells with HCMV (by centrifugal enhancement to synchronize infection, followed by several washing steps) and only 3h later (after the virus has been taken up) we add the stimuli (as above), and then analyze IE1 transcript levels 6h later. Here, in a setup where the virus was taken up by the cells prior to activation of the BMPR-IFNAR pathways, we clearly see an effect on IE1 expression as well (same phenotype as in Figure 3B). Hence, we conclude that the BMP/IFN stimulation affects HCMV on the level of de novo viral gene expression, and not at the level of virus uptake.

3) In figure 3A/B, the authors also suggest that the ability of BMP15 or Activin B, alone, to limit IE expression might be due to suppression of fibroblasts proliferation - do they have any evidence that this is the case as this seems counterintuitive to this reviewer on the basis that cells in S-phase or G2/M support IE expression poorly compared to cells in e.g. G0/G1?

Indeed, the effect of BMP15 and Activin B on HCMV is very interesting, and we are following up on this observation in a follow-up project since the aim of this study was to identify a BMP-type I IFN axis in the context of HCMV infection as already shown for other BMPs. Only BMP9 showed a co-stimulatory effect with IFN, and this is why we focused our attention on this BMP. To better explain our rationale for investigating the BMP9-IFN phenotype, we rephrased this paragraph in the results section of our manuscript.

4) Figure S1A, again, have the authors considered confirming protein levels for these receptors?

We have included a new panel now in this Figure showing protein expression of ALK1, ALK2, ALK3, BMPR2 in 293T. While ALK1 and BMPR2 are expressed very well, comparable to their expression level in HFF-1, ALK2 and ALK3 seem to be expressed at very low level. We could not find commercial antibodies for ACVR2A and ACRV2B, and also the BMP community did not have a recommendation to detect these receptors by immunoblotting.

5) Figure S1C - fold induction over what?

As for Main Figure 6, fold induction was calculated over unstimulated cells. We indicated this now more clearly in the corresponding legends for Figure 6 and S1C.

6) Figure 3D. I see no primary data that shows that antibodies to BMP4/6/15/Activin B block infected HHF supernatants from activating BRE-Luc - this should be shown or stated.

As elaborated above, our study focuses solely on BMP9 after we identified it as the only BMP having a co-stimulatory effect with IFN in the context of HCMV infection (shown in Figure 3B/C). The question behind Figure 3D was whether the response induced in the 293T reporter assay was conveyed by BMP9, and this is indeed the case. We do not claim, nor deny, that other BMP antibodies may reduce the luciferase fold induction shown in Figure 3D to a certain degree. However, we clearly show that the BMP9 antibody used for this experiment is highly specific for BMP9 only, and does not impact reactivity of BMP4, BMP6, BMP15 and Activin B (please see supplementary Figure 1D). Since the BMP9 antibody almost completely abolishes the response shown in Figure 3D, we are confident in the conclusion that the induced response is mainly attributed to BMP9 present in the supernatant.

7) Figure 6B. Overexpression of US18 or US20 (or together) dampens the ability of BMP9 to activate BRE-Luc. Have the authors confirmed that this is as a result of down-regulation of BMP receptors in their reporter cell line?

Fielding et al (Fielding et al. eLife 2017;6:e22206; DOI: 10.7554/eLife.22206) show that BMP receptors are downregulated from the cell surface of HCMV-infected cells (see our introduction), so this is the most likely mechanism for how US18/20 impair BRE-Luc activation. We are currently addressing how US18/US20 mechanistically mediate this downregulation, and/or whether there are effects on receptor signalling in addition to receptor downregulation.

Additionally, (figure 6B, lower panel) why does US20 (but not US18) augment IFNB activation of MX1-Luc (and what happens if both are co-expressed)? And, on this basis, is the statement that US18/US20 "...do not affect IFNAR signaling" (line 211) correct - or consistent with the view, throughout, that the absence of US20 increases sensitivity to IFNB?

We have adjusted our wording from "...do not affect IFNAR signalling" to "...US18 and US20 do not inhibit IFNAR signalling". As elaborated above (reviewer #1 regarding Figure 6B), since we do not carry along a control for upregulation of IFNAR signalling, the observed effect mediated by US20 on MX1 cannot be conclusively interpreted.

8) Figure 8. Whilst I accept the authors believe that the effects of US18 and US20 stop viruses are due to the inability of these viruses to down-regulate BMP receptors, have they ruled out additional effects such as decreased BMP9 induction - do effects on Luc expression in their BRE-Luc reporter cells differ if they use supernatants from WT/US18stop/US20stop virus?

This is a very interesting point, and opens up a new direction of the project that we think is outside of the scope of this manuscript, since it mainly focusses on the presentation of BMP9 as a novel regulator of the innate immune response to HCMV infection. We cannot exclude that infection with the mutant viruses lacking US18 and/or US20 may result in differences in BMP9 secretion. However, BMP9 secretion upon HCMV infection can be detected at very early time points point infection (Figure 3D) where the virus is still in the phase of taking over the control of cellular processes. Therefore, we would hypothesize that differences in BMP9 secretion at early stages of infection between WT are negligible. Nevertheless, this is an interesting point which we will address in a follow-up project which focuses on the determination of the mechanism of action that BMP9 uses to enhance the initial antiviral response to infection. Thank you for this insightful input.

9) Finally, Pham et al (2021) have shown that TGFβ, also induced upon lytic infection, limits induction of type I interferons. This should at least be mentioned.

Thank you for pointing this out to us, we have mentioned it now in the introduction.

Referee #3:

This study demonstrates that BMP9 plays a role in the antiviral response to HCMV, partly through interfacing with the IFN response, and that HCMV antagonizes this response through the US18 and US20 proteins. This information is interesting, and it is very clear that the authors have discovered some novel biology relating to BMP9 function and its antiviral activity, as well as identifying HCMV genes that antagonize these processes. This certainly impacts our understanding of HCMV, and may be relevant to other viruses too. However the study also has several limitations. My main two criticisms are that:

Thank you very much for your feedback and positive words.

1. The study is very correlative, and although many of these correlations are consistent with the hypothesis put forward, they stop short of actually proving the mechanistic links that they claim. *While we agree that we have not fully deciphered the exact mechanism how BMP9 regulates HCMV infection, we think that the importance of our paper lies in the combination of several novel findings: we show that BMP9 has antiviral activity against HCMV, that BMP9 protein is secreted upon viral infection, and that the virus encodes a specific defense mechanism to circumvent BMP9 signaling and thus evade repression. Therefore, we 'close the loop' – the best evidence that a potential antiviral pathway is physiologically relevant is that it is induced by a viral infection and that a virus takes steps to inhibit it. Our findings tick these boxes, and no other paper has done this yet for BMPs and so should be of general interest. We are currently addressing further precise mechanisms behind these novel findings, but these studies are longer term and are beyond the scope of this manuscript.*

2. The vast majority of the work uses a huge amount of biochemical and cell biology analysis to draw conclusions about the signalling pathways involved. What really matters at the end of the day is the effect of these pathways on the virus - i.e. do these pathways result in better viral control. And here the data could be far stronger - the use of IE1 or UL44 mRNA levels as a readout for 'antiviral capacity' is unusual, and (in my opinion) not sufficient.

We as virologists agree that more assays showing the effect on HCMV lytic life cycle progression would be great to have. However, we were faced with limitations in this regard. First, we would need to perform the assays (genome copy number measurements, plaque assays) in serum free medium, because BMP9 levels in the serum are too high and effects are negligible. This is not possible with HFF-1 since they start to be affected by starvation already after 10-12 hours. We cannot measure any other parameter than immediate early/early gene expression at this early time point since HCMV is a slowly replicating herpesvirus – we would need to starve and stimulate the cells with BMP9 for several days (at least 2-3 days for genome copy numbers, and even more so for plaque assays). We have previously worked with herpesviral antagonists (Stempel et al., 2019, Chan et al., 2016) or interferon-stimulated genes (Gonzalez-Perez, 2021) where phenotypes were only present in specific time windows post infection, but still had an impact on overall viral fitness. Thus, while we agree that more readouts have to be done to define the role of BMP9 in the innate antiviral response to HCMV infection, we show that BMP9/IFN signaling plays a role very early after infection, and using a later readout may not show significant differences since BMP9 enhances type I IFN signaling, but is not crucial for it.

My specific points in more detail are:

Fig 3B. At a MOI=0.5, how come all cells are infected? Would they see a stronger antiviral impact if they were infecting cells at a level just below 100%?

We thank the reviewer for pointing this out and do agree that the labeling of the figure is unhelpful. We aimed to say that we normalized the number of HCMV IE1+ cells to the total number of cells in the respective well, and displayed them normalized to the number of HCMV IE1+ cells in infected, but further untreated cells (the white column). We have adjusted the labeling of the figure and the corresponding figure legend.

Figure 5A - the effects of BMP9 on STAT1 phosphorylation are very weak, yet they claim that this is the likely mechanism by which BMP9 affects IFN signalling. Are those changes really sufficient to make that claim? Can they formally show that this conclusion is mechanistically true?

This experiment was performed three times independently and the bands were quantified, showing significant differences. Hence, we are convinced that these changes suffice to support our conclusions. Showing this mechanistically is very difficult and currently being studied in our lab. As mentioned above, we think that this question is beyond the scope of our study.

Fig 9. A key claim of the paper is that targeting of ALK1/2 by US18/US20 is responsible for the effects seen. Yet in the referenced paper, the level of ALK1/2 recovery in the delta US18/20 mutant looks very minimal, and the addition of BMP9 has clear effects against the WT virus, when (assuming this virus has targeted ALK1/2), it shouldn't be. Can the authors show by flow the levels of ALK1/2 in these mutants, at both early and late times, in order to support their conclusions? Is there a way to modulate ALK1/2 downregulation (e.g. by overexpression/knockout) to definitively show that this is how US18/20 are blocking BMP9 signalling?

We thank the review for this suggestion. As mentioned above, Fielding et al (Fielding et al. eLife 2017;6:e22206; DOI: 10.7554/eLife.22206) show that BMP receptors are downregulated from the cell surface of HCMV-infected cells, specifically that ALK1 and ALK2 expression can be rescued in the absence of US18 and US20. We have also thought of generating different ALK KO or use siRNA-mediated knockdown of specific receptors as well as analyze the surface expression of different receptors at different stages of HCMV infection. However, the intriguing part about BMPs is that, while they do have their preferred receptors to bind to, these relationships are not exclusive and there will always be a significant overlap of signals from overlapping subsets of receptors (i.e. reviewed in Mueller TD et al., FEBS Lett. doi: 10.1016/j.febslet.2012.02.043). Thus, while BMP9 may miss the main receptors in specific KO cells, a portion of BMP9 will still induce intracellular signaling by binding to other receptors, hence a completely abolished signal in ALK1/2 KOs upon BMP9 stimulation is not expected. While we agree that this is an important point, we think that the interplay of different BMP

receptors, their promiscuity and specificity need to be addressed in a future study and is outside of the scope of this manuscript.

Along related lines, the authors use a single stop codon in place of the start codon to knock out their genes - there are good reasons for choosing this modification, but it comes with the risk of residual translation. Can they show by western blot that this results in a complete knockout of US18/20 expression?

The use of stop viruses is a widely accepted method used in the field, and we decided against complete knockout mutants to ensure that we do not disrupt expression of other ORFs, i.e. US19, which is transcribed on the same multi-cistronic mRNA. Also, we do not have antibodies detecting US18/US20, and they are not available in the HCMV community to our knowledge since we asked for them on several occasions – and even then, detection of truncated proteins may not be possible with these antibodies. While we agree that there is a risk for the generation of truncated US18 and/or US20 proteins in these knockouts, we have double-checked with ribosome profiling data available for HCMV (Stern-Ginossar, 2012 Science and 2015 JVI) when generating these viruses. According to ribosome profiling data, other start codons present in the coding sequence are not employed and thus would either not lead to truncated proteins, or would result in proteins levels that are negligible.

Fig 9. I'd expect the ability of US18/20 to target BMP9 mediated antiviral effects to be strongest at late times, following de novo expression. In that case, why is the only analysis done at early times, when US18/20 levels delivered from the virion are likely to be fairly low? Furthermore, the data here seems inconsistent with some of the core claims, with the WT showing effects that are very similar to the knockout virus in some cases, and even an enhanced antiviral effect against the WT in some cases (e.g. the effect of BMP1 on IE1 mRNA). How can the authors explain these results if their hypothesis is true? The choice to measure RNA transcripts as a readout is odd, and gives very limited information - e.g. does this reflect lower expression in the cell, or fewer infected cells? What are the knock-on consequences of this to the virus? It's also odd to choose UL44 given that I'd assume UL44 is not expressed well at 6h? I think it would really help their case if the authors could also show antiviral effects at the level of virus spread or release - i.e. adding BMP9/IFN β onto cultures of spreading virus, to formally demonstrate that these molecules and pathways have the antiviral effects they claim, and that these activities are functionally antagonized by US18/20.

We thank the reviewer for these suggestions and will try to address them in future studies. However, there are limits to working with BMPs since they are multifunctional signaling factors.

(1) The suggestion of the reviewer to add BMP9 onto cultures of spreading virus may miss the phenotype that BMP9 is showing in our experiments. Cultures of spreading virus do not represent the initial antiviral responses induced by cells upon the first encounter with the virus – the supernatant of infected cells is loaded with a variety of cellular and viral cytokines which are simply not present upon initial infection, and which may modulate BMP9 activity. Thus, investigating the impact of BMP9 addition to fully infected cells may be difficult to interpret in comparison to our earlier timepoint experiments.

(2) Our reasoning to focus on RNA transcript readouts have been addressed in previous comments, and we will focus our efforts in the future to include various readouts to strengthen our claims. However, serum-starvation of HFF-1 is not possible for longer than a period of approx. 12 hours, and prolonged treatment with BMPs with or without infection will ultimately affect the cellular status to a degree that cannot be compared to freshly infected cells.

(3) While UL44 is a protein which is expressed with delayed-early kinetics, thus protein levels can (usually) be detected upon 24 hours and onwards, transcripts must be present prior to protein translation and are very well detectable at early stages of infection. Additionally, UL44 transcript levels follow the same phenotype as observed for IE1 transcripts – thus we do not agree with the reviewer that UL44 is an odd choice. Moreover, we would argue the other way around: By including UL44, we can show that the observed effect is not restricted to immediate-early gene expression, but a general impact on viral gene expression independent on the kinetic class.

Similar to the reviewer, we are very much interested in the processes affected by BMP9 and the antagonism by US18 and US20, but are at the same time limited to readouts that are currently possible with the cell line and its compatibility with BMP treatment. Surely, using other cell lines would be an option, however, primary HFF-1 cells are an established cell line in the HCMV field, alternatives are oftentimes immortalized cells which are fundamentally distorted in cell proliferation and migration – two of the main aspects of BMP signaling. Thus, we aimed to use primary cells and focused on the readouts that are currently possible and deliver results that can be used for interpretation.

Figure S2 - ideally virus needs to be purified on a glycerol/tartrate gradient before concluding that proteins are part of the virion, as opposed to cellular contamination. It would also help if a control for cellular contamination was included in the blots (e.g. calnexin?).

We agree that gradient purified virus would cast away the last doubt, but we have two arguments that rule out cellular contamination:

- 1. If this was cellular contamination, would there be an effect on BMP signaling? The viral proteins need to enter the cells – they need to be present in the intact virion to do this, they are membrane proteins and do not penetrate cells on their own.*
- 2. Bogdanow et al have published a beautiful cross-linking proteomics paper in which they show that US18 and US20 are indeed present in virions (<https://www.nature.com/articles/s41564-023-01433-8>, please see supplementary table 4).*

Dear Prof. Brinkmann,

Thank you for the submission of your revised manuscript to our editorial offices. I have now received the reports from the three referees that have been asked to re-evaluate your paper, you will find below. As you will see, the referees now support the publication of the study in EMBO reports. Referee #3 has some remaining concerns and suggestions to improve the study, I ask you to address in a final revised manuscript.

- Please provide your final manuscript text as a .docx formatted file without the figures included.
- I would like to avoid the brackets in the title. How about:
Novel role of bone morphogenetic protein BMP9 in innate host responses to HCMV infection
- Please remove the conflict of interest statement from the title page. We updated our journal's competing interests policy in January 2022 and request authors to consider both actual and perceived competing interests. Please review the policy <https://www.embopress.org/competing-interests> and update your competing interests if necessary. Please name this section 'Disclosure and Competing Interests Statement' and put it after the Acknowledgements section. Please also state there that you are an editorial board member of EMBO reports.
- Please add a paragraph titled 'Biosafety' to the methods section providing details on where and how biosafety-relevant experiments with viruses were performed and that these were approved, and by whom (institution, government).
- Please make sure the final manuscript text file contains a 'Data Availability Section'. Please state there that no large primary datasets have been generated and deposited for this study.
- Regarding data quantification and statistics, please make sure that in the final manuscript text file the number "n" for how many independent experiments were performed, their nature (biological versus technical replicates), the bars and error bars (e.g. SEM, SD) and the test used to calculate p-values is indicated in the respective figure legends (main and Appendix figures). Please also make sure that all the p-values are explained in the legend, and that these fit to those shown in the figure. Please provide statistical testing where applicable. Please avoid the phrase 'independent experiment', but clearly state if these were biological or technical replicates. Please also indicate (e.g. with n.s.) if testing was performed, but the differences are not significant. In case n=2 please show the data as separate datapoints or bars without error bars and statistics. See also: <http://www.embopress.org/page/journal/14693178/authorguide#statisticalanalysis>

If $n < 5$, please show single datapoints for diagrams.

For the diagram in Fig. 2A you state that the experiment was performed two independent times with two technical replicates each, and that one representative experiment is shown. I.e. please show here the data as 2 separate datapoints or bars without error bars.

Please provide statistics also for panels 2D and 2E.

For the diagram in Fig. 4E you state that the experiment was performed two independent times with two technical replicates each, and that one representative experiment is shown. I.e. please show here the data as 2 separate datapoints or bars without error bars.

For the data shown in panels EV1B-D please indicate how many (technical?) replicates are shown. In case $n \geq 3$, please add statistics. If $n=2$, then please show the data as 2 separate datapoints or bars without error bars.

- Please add a title to the legend for Fig. EV1.

- As I indicated in my first decision letter, we now request the publication of original source data with the aim of making primary data more accessible and transparent to the reader. It seems you have been contacted already by our source data coordinator, who indicated which figure panels we would need source data for. I attach again the source data checklist. Please make sure that all the requested source data is provided. Please upload all source data for one figure as one pdf per figure or (if there is more than one file) ZIPed together as one folder. Finally, please upload the filled in source data checklist with your final revised files.

In addition, I would need from you:

- a short, two-sentence summary of the manuscript (not more than 35 words).
- two to four short (!) bullet points highlighting the key findings of your study (two lines each).

- a schematic summary figure as separate file that provides a sketch of the major findings (not a data image) in jpeg or tiff format (with the exact width of 550 pixels and a height of not more than 400 pixels) that can be used as a visual synopsis on our website.

Best,

Referee #1:

The authors addressed most of my questions/concerns and I am satisfied with their explanations.

Referee #2:

This is a revised version of a manuscript that I previously reviewed and for which I requested a number of clarifications and additional information.

The authors have addressed all my points comprehensively with, in some cases, additional experimental data.

I am happy with the revisions.

Referee #3:

I thank the authors for their clear replies to my points. The majority of the points are fine, however there are two that they have not addressed, that (as far as I can tell) are relatively simple, and would ensure that the claims made are clear and evidence-backed. Those points are:

One of the key implications of their work is that US18/20 have the observed effects on BMP9 signalling because they modulate cell-surface levels of ALK1/2. I asked whether they could show this, by staining for ALK1/2 following infection of cells with the wildtype or mutant viruses. Can they show this data, for both virion-delivered and do-novo expressed protein? Whether or not they see effects does not change the fact that US18 and US20 have effects on BMP9-induced signalling, but it does change the interpretation of how these proteins mediate these effects.

Secondly, I asked whether the antiviral effects observed in Figure 9 on levels of UL44/IE1 mRNA are due to a reduction in the number of cells infected, or a reduction in per-cell transcript levels. Can they confirm, maybe via immunofluorescence? This speaks to whether the impact of this antiviral mechanism is strong enough to block infection (as IFN does), or if it just modulates the efficiency of viral transcription.

Institute of Genetics
Prof. Dr. Melanie M. Brinkmann

Technische Universität Braunschweig | Institute of Genetics
Spielmannstraße 7 | 38106 Braunschweig | Germany

To

The Editors of EMBO Reports

Technische Universität
Braunschweig

Institute of Genetics

Spielmannstraße 7
38106 Braunschweig
Germany

Prof. Dr. Melanie M. Brinkmann

phone: +49 (0) 177 37 57 315
m.brinkmann@tu-braunschweig.de
<http://tu-braunschweig.de/ifg>
<http://tinyurl.com/BrinkmannLab>

Kontonummer 19 99 200
BLZ 250 500 00

IBAN: DE79 2505 0000 0001 9992 00
BIC (Swift Code): NOLADE2H
USt.-ID-Nr.: DE 152 330 858
Steuer-Nr.: 14/20124509

Braunschweig, 3rd of January 2024

Dear Editors of EMBO Reports, dear Achim,

Today, we submit the second revision of our manuscript "Novel role of bone morphogenetic protein 9 in innate host responses to HCMV infection".

Addressing the comments of reviewer #3, we now include an immunoblot as EV3 showing equal expression of the tegument protein UL82/pp71 three hours after infection with WT and mutant virus, demonstrating that the infection efficiency of the two viruses is comparable and that viral entry is not impaired. We were not able to provide flow cytometry analysis of cell surface BMP receptor expression since the commercial antibodies we tested did not give interpretable results in HFF-1 cells. However, Fielding et al have already convincingly demonstrated by plasma membrane proteomics that BMP receptors are downregulated in HCMV-infected HFF-1 cells.

We have also addressed all editorial requests.

We are looking forward to your decision.

Yours sincerely,

Prof. Melanie Brinkmann
Technische Universität Braunschweig
Institute of Genetics
Spielmannstrasse 7
Braunschweig, Niedersachsen 38106
Germany

Dear Prof. Brinkmann,

I am very pleased to accept your manuscript for publication in the next available issue of EMBO reports. Thank you for your contribution to our journal.

Yours sincerely,
